# Transformers for Mixed-type Event Sequences

**Felix Draxler, Yang Meng, Kai Nelson**
University of California, Irvine
{fdraxler,mengy13,ktnelson}@uci.edu

**Lukas Laskowski**
Hasso Plattner Institute, University of Potsdam
lukas.laskowski@hpi.de

**Yibo Yang, Theofanis Karaletsos**
Chan-Zuckerberg Initiative
{yyang,tkaraletsos}@chanzuckerberg.com

**Stephan Mandt**
University of California, Irvine
mandt@uci.edu

## Abstract

Event sequences appear widely in domains such as medicine, finance, and remote sensing, yet modeling them is challenging due to their heterogeneity: sequences often contain multiple event types with diverse structures—for example, electronic health records that mix discrete events like medical procedures with continuous lab measurements. Existing approaches either tokenize all entries, violating natural inductive biases, or ignore parts of the data to enforce a consistent structure. In this work, we propose a simple yet powerful Marked Temporal Point Process (MTPP) framework for modeling event sequences with flexible structure, using a single unified model. Our approach employs a single autoregressive transformer with discrete and continuous prediction heads, capable of modeling variable-length, mixed-type event sequences. The continuous head leverages an expressive normalizing flow to model continuous event attributes, avoiding the numerical integration required for inter-event times in most competing methods. Empirically, our model excels on both discrete-only and mixed-type sequences, improving prediction quality and enabling interpretable uncertainty quantification. We make our code public at https://github.com/czi-ai/FlexTPP.

## 1 Introduction

Time series are a ubiquitous data modality, arising in a wide range of domains such as electronic health records [Wornow et al., 2023], high-frequency trading [Bacry et al., 2015], click stream prediction [Gündüz and Özsu, 2003], hardware and access logs in cybersecurity [Fortino et al., 2023], and earthquake monitoring in remote sensing [Ogata, 1998]. Particularly common are time series with irregular time intervals between adjacent events (e.g., lab test times), where each event is associated with metadata known as *marks* (e.g., test results). Such time series are known as *Marked Temporal Point Processes* (MTPPs) [Daley and Vere-Jones, 2008].

Despite their broad applicability, existing MTPP models typically handle only simplified forms of real-world event data. Prior work focuses on discrete marks such as event types [Mei and Eisner, 2017] or sets of items [Chang et al., 2024], or on continuous marks such as spatio-temporal features [Chen et al., 2021, Dong et al., 2024]. In contrast, many real datasets include mixed-type metadata that combines discrete attributes (e.g., diagnosis codes, transaction types) with structured continuous values (e.g., measurements, durations). The length and structure of this metadata also vary across events—for instance, due to different medical test panels or variable-length action logs in cybersecurity.

Ignoring these heterogeneities leads to a mismatch between model assumptions and real data. Recently, Event Stream GPT [McDermott et al., 2023] modeled heterogeneous event structures, but it treats continuous marks with unimodal Gaussians, limiting expressivity and autoregressive sequence

39th Conference on Neural Information Processing Systems (NeurIPS 2025).

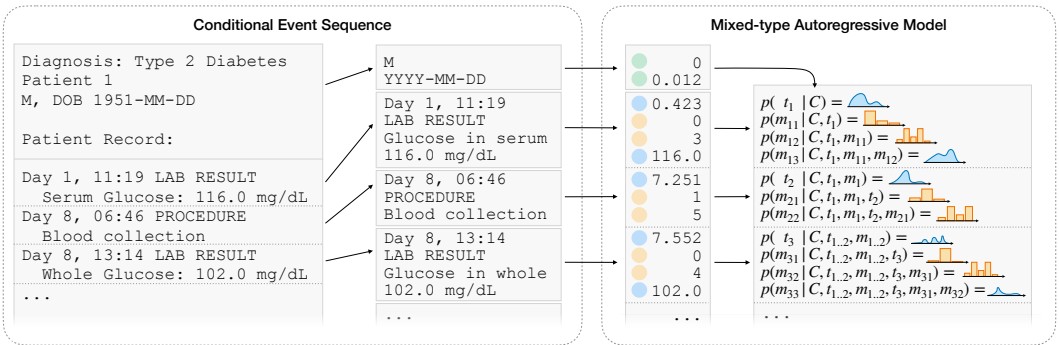

Figure 1: We propose a transformer-based MTPP model for event sequences of varying length and structure. *(Left)* The example shows how we model electronic health record as a series of events with variable-length and and mixed discrete and continuous variables, conditioned on demographic information. *(Right)* We fit the point process using a conditional autoregressive model with separate output heads for discrete and continuous entries.

generation. To address this, we propose a streamlined MTPP architecture that leverages expressive distributions for all variable types, yielding a unified and powerful joint probabilistic model. As illustrated in Figure 1, our approach converts each event sequence into a single mixed-type vector and models it with a Transformer-based autoregressive network. Discrete event types act as "headers" that determine variable event structure, and separate continuous and discrete prediction heads capture heterogeneous mark attributes.

With this more expressive mark space, we can fully exploit conditioning MTPPs on external input. While prior work considered limited forms of conditioning—such as static covariates—with minimal impact on prediction [Šeputis et al., 2022, Verheugd et al., 2020, Isik et al., 2023], we treat conditioning as central, addressing problems that cannot be solved without it. This broader perspective allows us to use conditional MTPPs as a flexible probabilistic framework for structural mixed-type regression.

Together, our contributions are as follows:

- We propose FLEXTPP, a flexible Transformer-based MTPP framework that supports variable-length, mixed-type marks, encompassing both discrete and continuous event attributes (section 4.1). Beyond standard generative modeling, we treat MTPPs as a structured prediction framework, conditioning event sequences on auxiliary input to solve regression and prediction tasks (section 4.2).

- FLEXTPP is intensity-free, avoiding the numerical integration required by competing intensity-based approaches. Empirically, it also outperforms both intensity-free and intensity-based methods on the discrete-only EasyTPP benchmark [Xue et al., 2024] (section 5.1).

- We demonstrate how FLEXTPP's flexibility can be leveraged in practice: modeling heterogeneous events in electronic health records, performing event-series annotation with uncertainty quantification and extracting event dependency structure (sections 5.2 to 5.4).

Together, we propose a simple and versatile framework to model heterogeneous event sequences.

## 2 Related Work

There is extensive literature on modeling marked temporal point processes (MTPPs); we summarize our work in relation to existing methods in Table 1, and give a more detailed discussion below.

The predominant line of work models MTPPs with so-called Hawkes processes [Hawkes, 1971], parameterizing an *intensity function* that captures the rate at which the next event will occur. Several neural realizations have been proposed, first using recurrent neural networks [Du et al., 2016, Mei and Eisner, 2017] and later Transformers [Zhang et al., 2020, Zuo et al., 2020, Yang et al., 2021],

Table 1: Overview of related MTPP methods in the literature. To the best of our knowledge, we are the first to model expressive distributions to events with mixed data types and varying dimensions. In addition, we consider conditional MTPPs, where the point process depends on auxiliary input. [*] collects [Du et al., 2016, Mei and Eisner, 2017, Zhang et al., 2020, Zuo et al., 2020, Yang et al., 2021, Zhuzhel et al., 2024, Chang et al., 2025, Gao et al., 2024, Xu and Zha, 2017, Liu and Quan, 2024, Isik et al., 2023].

| | Data types | | Mark dimension | | Conditional |
|---|---|---|---|---|---|
| | Discrete | Continuous | Multiple | Variable | |
| Classical Hawkes Process [Hawkes, 1971] | ✓ | ✗ | ✗ | ✗ | ✗ |
| Neural Hawkes Process [*] | ✓ | ✗ | ✗ | ✗ | ✓ |
| Instensity-Free TPP [Omi et al., 2019, Shchur et al., 2020] | ✓ | ✗ | ✗ | ✗ | ✗ |
| Conditional event generators [Dong et al., 2024] | ✗ | ✓ | ✓ | ✗ | ✗ |
| Set-valued MTPPs [Chang et al., 2024] | ✓ | ✗ | ✓ | ✓ | ✗ |
| Neural spatio-temporal process [Chen et al., 2021] | ✗ | ✓ | ✓ | ✗ | ✗ |
| Event Stream GPT [McDermott et al., 2023] | ✓ | ✓/✗ | ✓ | ✓ | ✓ |
| Ours | ✓ | ✓ | ✓ | ✓ | ✓ |

convolutional networks [Zhuzhel et al., 2024], and state-space models [Chang et al., 2025, Gao et al., 2024]. Other notable directions model the next-time distribution instead of intensities [Omi et al., 2019, Shchur et al., 2020], consider mixtures of series [Xu and Zha, 2017], or leverage the inductive biases of pretrained language models Liu and Quan [2024]. While the aforementioned methods only consider MTPPs where each event is associated with one discrete mark (the event type), other kinds of marks have also been considered. Chen et al. [2021] model spatio-temporal point processes using Neural ODEs, where marks are purely continuous. Similarly, Dong et al. [2024] model multi-dimensional continuous marks using joint generative models such as diffusion models. Chang et al. [2024] model sets of discrete marks. Similar to our approach, McDermott et al. [2023] propose a system to learn heterogeneous event structures, but they do not learn an expressive model for continuous event data, limiting performance. We generalize the previous work into one probabilistic framework that jointly handles variable-length and mixed-type mark spaces.

Some work has considered conditioning MTPPs on external input such as for predicting failures of water pipes [Verheugd et al., 2020] or in medical context [Šeputis et al., 2022, Isik et al., 2023]. Our Transformer-based architecture allows for natural conditioning, which we exploit both for feeding covariates and as the input to regression tasks.

In the context of the related work, our model is Transformer-based, intensity-free, and allows for variable-length and mixed-type marks.

In generative modeling, autoregressively predicting variable-length sequences is at the core of language models such as GPT [Mikolov et al., 2010, Radford et al., 2018]. Mixed data types are often integrated in the input but cannot be generated, for example, in vision-understanding models such as Flamingo [Alayrac et al., 2022]. Jointly modeling mixed modalities has been achieved by discretizing (tokenizing) continuous values or fusing other generative models and language models. We refer to [Xu et al., 2023] for a comprehensive overview. Similar to our modeling approach, [Fakoor et al., 2020] propose and [Strauss and Oliva, 2021] perform joint modeling of discrete and continuous data with a transformer backbone, inspired by autoregressive models for continuous data [Germain et al., 2015]. However, neither of these works exploit the full flexibility of modeling variable-length, mixed-type data.

## 3 Background: Marked Temporal Point Process

A natural way to describe the distribution of event sequences over time is through a *Marked Temporal Point Process (MTPP)*. An MTPP is a stochastic process describing a sequence of events $\{(t_i, m_i)\}_{i=1}^T$, where each event is characterized by its occurrence time $t_i \in \mathbb{R}_+$ and an associated mark $m_i \in \mathcal{M}$. While $\mathcal{M}$ traditionally describes a single discrete or continuous variable (see section 2 and table 1), we generalize it in section 4.1 to mixed discrete–continuous spaces. Event sequences are inherently stochastic: both the timing and the mark of the next event are inherently

uncertain. Their joint distribution can be factorized using the chain rule as:

$$p(\{(t_i, m_i)\}_{i=1}^T) = \prod_{i=1}^{T} p(t_i, m_i \mid \mathcal{H}_{t_i}), \tag{1}$$

where $\mathcal{H}_{t_i} = \{(t_j, m_j) \mid t_j < t_i\}$ denotes the history of events up to $t_i$, capturing both temporal and mark dependencies.

Ignoring the marks for a minute (we will re-introduce them in section 4.1), a key design choice is how to model the one-dimensional distribution of the next event time $p(t_i \mid \mathcal{H}_{t_i})$.

First, *intensity-based* methods model the time distribution via a learned intensity function $\lambda(t_i \mid \mathcal{H}_{t_i})$, representing the instantaneous rate of events at a time $t$ [Daley and Vere-Jones, 2008]. Formally, the intensity function $\lambda(t \mid \mathcal{H}_t)$ defines the infinitesimal expected event rate, linking the point process to its conditional likelihood via integration over time:

$$p(t_i \mid \mathcal{H}_{t_i}) = \lambda(t_i \mid \mathcal{H}_{t_i}, [m_i]) \exp\left(-\int_{t_{i-1}}^{t_i} \lambda(t_i \mid \mathcal{H}_{t_i}) \, dt\right). \tag{2}$$

Since the intensity function can be any positive function, this formulation yields highly flexible time distributions. However, the numerical integration can be unstable and be expensive both at training (likelihood evaluation) and inference (sampling), depending on the modeling of the intensity function.

Second, *intensity-free* methods avoid numerical integration by directly modeling event times using parametric distributions $p_\varphi(t_i \mid \mathcal{H}_{t_i})$ conditioned on the history. Common choices are mixture models or one-dimensional normalizing flows, with parameters learned as a function of past events [Shchur et al., 2020]. This allows efficient evaluation and sampling.

Our proposed framework in section 4 is intensity-free. Outperforming previous models in section 5.1, it overcomes concerns about expressivity of intensity-free models [Chang et al., 2025].

# 4 FlexTPP

## 4.1 Variable-length, mixed-discrete-continuous events

Real-world events often carry heterogeneous information: some attributes are discrete (e.g., event type), others are continuous (e.g., laboratory test results such as glucose levels), and their number may vary depending on the event. To model such data, we introduce a unified mixed-modality framework that extends MTPPs to variable-length, mixed-type marks.

We are given events $i$ that consist of an arrival time $t_i$ and an associated mark $m_i$ that may contain a variable number of discrete and continuous components. Instead of treating these parts separately, we flatten the entire event sequence into a single sequence of scalar values $X = (X_1, \ldots, X_L)$, together with a corresponding type vector $D = (D_1, \ldots, D_L)$ indicating whether each entry is continuous or discrete.

Figure 1 illustrates this procedure. Formally, for each event $i$, the first mark $m_i^{\text{type}}$ denotes its type (always discrete). The remaining components $m_{i1..N_i}^{\text{add}}$ contain additional attributes whose number $N_i$ and data types $d_{ij} \in \{\text{cont}, \text{disc}\}$ depend on the event type. Formally,

$$X = (t_1, m_1^{\text{type}}, m_{11}^{\text{add}}, \ldots, m_{1N_1}^{\text{add}}, t_2, \ldots), \quad D = (\text{cont}, \text{disc}, d_{11}, \ldots, d_{1N_1}, \text{cont}, \ldots). \tag{3}$$

This vectorization yields a single sequence of length $L = 2T + \sum_{i=1}^T N_i$.

Modeling this vectorized event sequence is straightforward with an autoregressive model: The joint distribution of the sequence is $p_D(X_1, \ldots, X_L) = \prod_{l=1}^{L} p_{D_l}(X_l \mid X_{<l})$, where each conditional distribution matches the data type:

$$p_{D_l}(X_l \mid X_{<l}) = \begin{cases} \text{Cat}(X_l \mid X_{<l}) & \text{if } D_l = \text{disc}, \\ p_{\text{density}}(X_l \mid X_{<l}) & \text{if } D_l = \text{cont}. \end{cases} \tag{4}$$

Here, $\text{Cat}$ is a categorical distribution and $p_{\text{density}}$ a continuous density.

**Algorithm 1** Sampling from Mixed-Type Autoregressive Model
___
1: **Input:** Lengths $N(m^{\text{type}})$ and types $d(m^{\text{type}})$ for each event type $m^{\text{type}} = 1, \ldots, M$.
2: Initialize empty sequence $X = ()$, $i = 1$, $l = 1$.
3: **while** true **do**
4:     Sample time $X_l = t_i \sim p_{\text{density}}(X_l | X_1, \ldots, X_{l-1})$.
5:     Sample event type $X_{l+1} = m_i^{\text{type}} \sim \text{Cat}(X_{l+1} | X_1, \ldots, X_l)$.
6:     **if** $m_i^{\text{type}} = \texttt{EOS}$ **then**
7:         break.
8:     **end if**
9:     **for** $j = 1, \ldots, N(m_i^{\text{type}})$ **do**
10:         $X_{l+j+1} = m_{ij^{\text{add}}} \sim \begin{cases} \text{Cat}(X_{l+j+1} | X_1, \ldots, X_{l+j}) & \text{if } d(m_i^{\text{type}})_j = \text{disc} \\ p_{\text{density}}(X_{l+j+1} | X_1, \ldots, X_{l+j}) & \text{else.} \end{cases}$
11:     **end for**
12:     Update indices: $l = l + 2 + |G_{m_{i1}}|$ and $i = i + 1$.
13: **end while**
14: **return** $(t_i, m_i^{\text{type}}, m_i^{\text{add}})_i$
___

The above approach is a unified autoregressive model of variable-length, mixed-type MTPPs with parameters $\theta$. We train it by minimizing the negative loglikelihood of a dataset $\mathcal{D}$ of time series:

$$\min_{\theta} - \sum_{(X,D) \in \mathcal{D}} \log p_{D,\theta}(X). \tag{5}$$

Algorithm 1 shows how to sample from this model. Each event begins by sampling a continuous value for the arrival time, followed by an event type that determines the structure of the remaining entries. Sampling a special end-of-sequence token $\texttt{EOS}$ terminates the sampling.

## 4.2 Conditional Marked Temporal Point Processes

The general-purpose structure of our extension of MTPPs to variable data types makes them a natural choice for the output format in supervised prediction tasks. For example, we will later formulate the detection of events in a time series as a conditional marked time point process (see section 5.3). Similarly, for health records, feeding demographic information can increase modeling accuracy (see section 5.2). To this end, we modify the setup in section 4.1 to include conditional input $C$, so that the likelihood of each sequence is conditioned on $C$: $p_D(X|C) = \prod_{l=1}^{L} p_{D_l}(X_l | C, X_1, \ldots, X_{l-1})$.

Now that we have generalized MTPPs to handle heterogeneous mark data and condition on auxiliary information, we next instantiate this modeling approach with a Transformer architecture.

## 4.3 Flexible Marked Temporal Point Process (FLEXTPP)

We implement the the conditional likelihood $p_{D_l}(X_l | X_{<l})$ in eq. (4) with an autoregressive model. The idea is to map the history $X_{1\ldots i-1}$ to a fixed-dimensional representation:

$$\phi_l = \phi(D_l; C, X_1 \ldots X_{l-1}) \in \mathbb{R}^{d_m}, \tag{6}$$

which is then used to parameterize the conditional distribution of $X_l$. As illustrated in Figure 2, we implement the autoregressive computation with a Transformer [Vaswani et al., 2017]. This architecture captures long-range dependencies between events [Zhang et al., 2020, Yang et al., 2021, Zuo et al., 2020], and lets the size of intermediate representations grow naturally with the dimensionality of the marks.

For predicting discrete dimensions, we map $\phi_l$ into a categorical distribution as in language modeling,

$$\text{Cat}(x; \phi_l) = [\text{softmax}(\varphi_{\text{disc}}(\phi_l))]_x, \tag{7}$$

where we turn $\phi_i$ into logits $\varphi_{\text{disc}}(\phi_l)$ with a small fully-connected neural network.

For the continuous distributions, we map $\phi_i$ to the parameters $\varphi_{\text{cont}}(\phi_i)$ of a one-dimensional normalizing flow, as is common for continuous autoregressive models [Germain et al., 2015]. We use

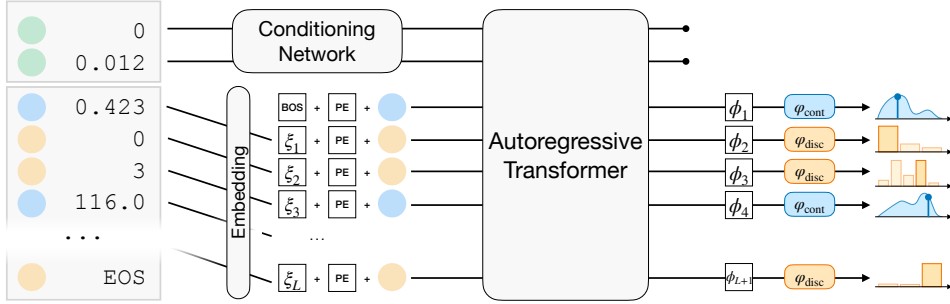

Figure 2: Our Transformer backbone naturally encompasses mixed-type marks and conditions. The vectorized input data (see fig. 1) is shifted by one position so that the embedding $\phi_l$ for predicting $X_l$ depends only on the history $X_{1,...,l-1}$ and the condition. The Transformer is equipped with a causal mask. We use a classification head for discrete dimensions, and a one-dimensional normalizing flow for continuous dimensions.

a rational-quadratic spline $z = f_{\varphi_{\text{cont}}}(x)$ [Durkan et al., 2019]. The flow defines a one-dimensional (conditional) density via the change of variables equation:

$$p_{\text{density}}(x; \phi_l) = \mathcal{N}(z = f_{\varphi_{\text{cont}}(\phi_l)}(x); m = 0, \sigma = 1)|f'_{\varphi}(x)|. \tag{8}$$

Note that for simplicity, we use eq. (8) both for the time and continuous mark dimensions ("intensity-free", see section 3). We call the above model Flexible Marked Temporal Point Process FLEXTPP when no condition is present, and we refer to FLEXTPP-C whenever it is conditioned.

## 5  Experiments

We evaluate our generalized MTPP framework in practice. We first confirm that FLEXTPP(-C) reliably works with discrete mark spaces on the EasyTPP benchmark [Xue et al., 2024] in section 5.1. We then demonstrate in section 5.2 how our generalized mark structure improves predicting medical procedures in electronic health records extracted from the EHRSHOT dataset [Wornow et al., 2023]. Finally, we propose to use MTPPs to make structured predictions in annotating time series in section 5.3. This MTPP heavily relies on its condition and can predict several types of annotations together with their structurally different properties.

For the experiments that concern generalized mark spaces, we compare the following versions of our model:

Conditional + discrete event type mark:   $p(t_1, m_1^{\text{type}}, \cancel{m_1^{\text{add}}}, \ldots, t_T, m_T^{\text{type}}, \cancel{m_T^{\text{add}}}|C),$   (9)

Unconditional + full mark (FLEXTPP):   $p(t_1, m_1^{\text{type}}, m_1^{\text{add}}, \ldots, t_T, m_T^{\text{type}}, m_T^{\text{add}}|\cancel{C}),$   (10)

Conditional + full mark (FLEXTPP-C):   $p(t_1, m_1^{\text{type}}, m_1^{\text{add}}, \ldots, t_T, m_T^{\text{type}}, m_T^{\text{add}}|C),$   (11)

The first model in eq. (9) only models time and a discrete event type as a mark, corresponding to how the bulk of the literature on MTPPs would model the data. We give these models access to the condition to allow for a fair comparison to the most general variant, FLEXTPP-C. The second model in eq. (10), in contrast, does not have access to the condition and only models the marginal marked time point process. Here, we measure how much the condition helps in making a prediction. Finally, our model in eq. (11) accepts a condition and jointly models all discrete and continuous marks.

We evaluate all models on a held-out test set in terms of negative log-likelihood (NLL), capturing the generative quality and the uncertainty in making predictions.

To compare our conditional model to the unconditional variant, we compute the negative logarithms of eqs. (10) and (11). Intuitively, a lower value means that the conditional model can make use of the condition to more accurately model the MTPP.

Comparing these likelihoods to the ones of the model that only captures discrete event types is not directly possible, as eq. (9) does not model the additional mark dimensions. We therefore also report

Table 2: Our model sets a new SOTA in four out of five datasets in terms of Negative Log-Likelihoods (lower is better, ↓) on EasyTPP Datasets [Xue et al., 2024], notably outperforming both intensity-free TPP [Shchur et al., 2020] and intensity-based methods. Marks are single discrete event type. Baseline values are due to [Chang et al., 2025].

| Model | Amazon | Retweet | Taxi | Taobao | StackOverflow |
|---|---|---|---|---|---|
| RMTPP [Du et al., 2016] | 2.136 (0.003) | 7.098 (0.217) | -0.346 (0.002) | -1.003 (0.004) | 2.480 (0.019) |
| NHP [Mei and Eisner, 2017] | -0.129 (0.012) | 6.348 (0.000) | -0.514 (0.004) | -1.157 (0.004) | 2.241 (0.002) |
| SAHP [Zhang et al., 2020] | 2.074 (0.029) | 6.708 (0.029) | -0.298 (0.057) | 1.646 (0.083) | 2.341 (0.058) |
| THP [Zuo et al., 2020] | 2.096 (0.002) | 6.659 (0.007) | -0.372 (0.002) | 1.712 (0.011) | 2.338 (0.014) |
| AttNHP[Yang et al., 2021] | -0.484 (0.077) | 6.499 (0.028) | -0.493 (0.009) | -1.259 (0.022) | 2.194 (0.016) |
| IFTPP [Shchur et al., 2020] | -0.496 (0.002) | 10.344 (0.016) | -0.453 (0.002) | -1.318 (0.017) | 2.233 (0.009) |
| MHP [Gao et al., 2024] | -0.496 (0.002) | 10.344 (0.016) | -0.453 (0.002) | -1.318 (0.017) | 2.233 (0.009) |
| S2P2 [Chang et al., 2025] | **-0.781** (0.011) | 6.365 (0.003) | -0.522 (0.004) | -1.304 (0.039) | 2.163 (0.009) |
| FLEXTPP (Ours) | -0.633 (0.039) | **5.646** (0.070) | **-0.763** (0.005) | **-1.402** (0.013) | **2.133** (0.004) |

the negative log-likelihood *of the time and the event type mark dimensions* under each model:

$$-\sum_{i=1}^{T} \log p(t_i, m_i^{\text{type}} | [C], m_{1..i-1}^{\text{type}}, [m_{1..i-1}^{\text{add}}]). \tag{12}$$

We pass in the additional mark dimensions and the condition only if the model version accepts them. A smaller loss means that the model can better estimate the arrival time and type of the next event when it has seen all marks of the previous event instead of just arrival times and marks. For example, in the clinical settings, it measures whether the next medical procedure prediction improves if we know the continuous lab results.

We give all details to replicate our experiments in appendix A.

## 5.1 EasyTPP Datasets

The EasyTPP benchmark [Xue et al., 2024] collects five datasets to compare models fitting MTPPs. All datasets have a discrete mark space modeling event types only, see appendix A.1. Table 2 shows that our intensity-free, Transformer-based MTPP outperforms both intensity-based and previous intensity-free methods.

This is an important data point in the MTPP modeling space, as it shows that intensity-free methods can perform well, all while avoiding numerical integration at training and inference time. In the next sections, we generalize the mark space beyond discrete marks.

## 5.2 Patient Record Data

As a first generalized experiment, we model a subset of EHR data from the EHRSHOT benchmark [Wornow et al., 2023]. Unlike standard EHR setups that capture only discrete events like procedures [Chang et al., 2025], our construction also includes continuous lab results, enabling a richer, mixed-modality representation of patient trajectories. This structure improves next-event prediction and enhances uncertainty quantification by integrating diverse clinical signals. See appendix A.2 for details.

To compile the dataset, we subset the patients diagnosed with 20 different diseases such as type 2 diabetes, dyspnea, atrial fibrillation, etc., to create disease-specific longitudinal analyses. For each disease, we capture the most used Current Procedural Terminology (CPT-4) codes that denote medical services and the most common procedures as discrete events, and model disease-related lab results as continuous events. Additionally, demographic data are incorporated as conditional inputs, enabling a better modeling of patient-specific dynamics.

Table 3 demonstrates the effectiveness of our proposed method. Besides the baseline in eq. (9) ("Procedures") which only models procedures as discrete events, we also compare FLEXTPP and FLEXTPP-C to another baseline ("+ Lab Tests") which additionally measures discrete lab test types but without their corresponding continuous results. By jointly modeling continuous lab results alongside discrete procedure types and conditioning on demographic information, our FLEXTPP-C approach achieves the best performance on both measures of negative log-likelihood ("Full" and

Table 3: By incorporating continuous lab results via eq. (11), our flexible models outperform baselines in predicting time and type of medical procedures: An MTPP with discrete marks would either not model lab events at all [Chang et al., 2025] ("Procedures"), or model that a lab test exists, but ignore its value ("+ Lab Tests"). Conditioning on demographic covariates (our FLEXTPP-C) always improves prediction quality for procedures and in the majority of cases for all events. Standard deviations in tables 9 and 10.

| Dataset | Time + Discrete Procedure NLL (↓) | | | | | | Full NLL (↓) | | | |
|---|---|---|---|---|---|---|---|---|---|---|
| | Procedures | + Lab Tests | ESGPT | ESGPT-C | FLEXTPP | FLEXTPP-C | ESGPT | ESGPT-C | FLEXTPP | FLEXTPP-C |
| Type 2 diabetes mellitus | 0.691 | 0.202 | 0.186 | 0.182 | 0.147 | **0.143** | 0.754 | 0.758 | 0.641 | **0.638** |
| Transplanted kidney | 0.755 | 0.170 | 0.199 | 0.192 | 0.134 | **0.130** | 0.859 | 0.863 | 0.670 | **0.668** |
| Transplanted lung | 0.912 | 0.089 | 0.078 | 0.075 | 0.068 | **0.063** | 0.851 | 0.857 | 0.581 | **0.546** |
| Dyspnea | 0.681 | 0.162 | 0.162 | 0.155 | 0.121 | **0.117** | 0.560 | 0.549 | 0.463 | **0.438** |
| Atrial fibrillation | 0.715 | 0.081 | 0.075 | 0.071 | 0.062 | **0.059** | 0.388 | 0.376 | 0.204 | **0.192** |
| Cardiac transplant disorder | 0.887 | 0.033 | 0.034 | 0.033 | 0.029 | **0.027** | 0.552 | 0.594 | 0.408 | **0.393** |
| End-stage renal disease | 0.697 | 0.166 | 0.169 | 0.164 | 0.128 | **0.117** | 0.757 | 0.753 | 0.580 | **0.578** |
| Transplanted heart | 0.869 | 0.041 | 0.033 | 0.032 | 0.029 | **0.028** | 0.736 | 0.701 | 0.546 | **0.521** |
| Congestive heart failure | 0.704 | 0.134 | 0.130 | 0.128 | 0.103 | **0.100** | 0.846 | 0.861 | 0.525 | **0.509** |
| Chronic pain | 0.735 | 0.099 | 0.095 | 0.093 | 0.076 | **0.073** | 0.765 | 0.743 | **0.162** | 0.165 |
| Neoplasm of female breast | 0.716 | 0.043 | 0.040 | 0.038 | 0.032 | **0.030** | 0.608 | 0.608 | 0.516 | **0.461** |
| Obstructive sleep apnea | 0.695 | 0.093 | 0.090 | 0.089 | 0.061 | **0.059** | 0.710 | 0.676 | 0.195 | **0.187** |
| Diabetes with complication | 0.718 | 0.176 | 0.164 | 0.160 | 0.133 | **0.129** | 0.694 | 0.703 | **0.579** | 0.582 |
| Anemia | 0.655 | 0.164 | 0.165 | 0.158 | 0.127 | **0.126** | 0.322 | 0.296 | **0.238** | 0.245 |
| Coronary artery disease | 0.692 | 0.144 | 0.138 | 0.134 | 0.108 | **0.105** | 0.966 | 0.986 | 0.636 | **0.625** |
| Hypothyroidism | 0.735 | 0.224 | 0.212 | 0.208 | 0.166 | **0.159** | 0.666 | 0.664 | **0.524** | 0.528 |
| Acute myeloid leukemia | 0.729 | 0.182 | 0.160 | 0.154 | 0.124 | **0.117** | 0.727 | 0.735 | 0.567 | **0.561** |
| Depressive disorder | 0.683 | 0.136 | 0.135 | 0.129 | 0.108 | **0.105** | 0.813 | 0.826 | 0.608 | **0.545** |
| Transplanted liver | 0.799 | 0.183 | 0.192 | 0.186 | 0.144 | **0.142** | 0.907 | 0.902 | 0.727 | **0.725** |
| Acute kidney injury | 0.645 | 0.201 | 0.191 | 0.185 | 0.144 | **0.139** | 0.706 | 0.711 | 0.621 | **0.605** |

"Time + Discrete Procedure"). ESGPT and ESGPT-C (McDermott et al. [2023]) are also implemented as additional baselines, where continuous lab results and arrival times are modeled by Gaussian and Log-Normal mixture heads, respectively, instead of the normalizing flow head used in our proposed methods. These results validate the advantage of incorporating continuous event types, patient-specific conditioning, and the expressiveness in modeling continuous attributes using normalizing flow in multimodal health record modeling.

## 5.3 Time Series Annotation

Finding eventful subsequences in time series data is crucial in many real-world applications. For example, in healthcare, timely and accurate identification of critical events can significantly impact diagnosis and treatment, and in audio annotation, where a continuous time series is transformed into an interpretable description.

A traditional approach is to have a classifier $p(m^{\text{type}}|t, C)$ predict the presence of an event type $m^{\text{type}}$ at a given time $t$ in an input sequence $C$ [Zhao et al., 2017]. However, fig. 3 (top right) illustrates how this is fundamentally limited when several events are present at the same time: It expresses uncertainty in the event class instead of predicting both classes to be present. Another restriction is that classifiers are often evaluated on small windows to save compute for long input time series.

We lift these restrictions by directly predicting the target sequence of events as a MTPP under our flexible framework FLEXTPP-C. This allows for parallel events, and compute is naturally bounded by the number of events. In this setting, each event is characterized by a start timestamp (continuous), a type label (discrete), a duration (continuous), and additional event data (mixed length, mixed type). Because our framework is fully probabilistic, we get uncertainties for all these quantities.

We evaluate our model on a set of synthetic input time series. Our dataset is based on a noisy harmonic oscillator, a common model in biology, physics, acoustic and mechanical systems. It is given by the following system of stochastic differential equations:

$$dx = vdt + \sigma_x dw_x, \qquad dv = (-\gamma v - \omega^2 x + f(t))dt + \sigma_v dw_v. \tag{13}$$

For each time series in our dataset, we first choose base values for the constants $\omega, \gamma, \sigma_x, \sigma_v$. We then randomly sample a list of events that alter the dynamics over some amount of time by changing one or several of the constants, or add an external force $f(t)$. See appendix A.3 for details.

Figure 3 shows a typical example time series with annotated events. Our FLEXTPP-C predicts event times and properties, together with uncertainty intervals. Note how an MTPP with discrete marks only captures the start times and type of events. Table 4 evaluates eqs. (9) and (11) to compare negative log-likelihoods. We also train a sliding window classifier [Zhao et al., 2017] with a finite

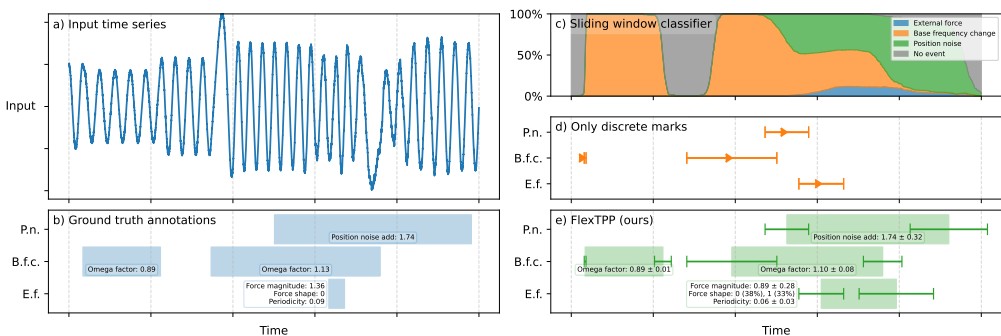

Figure 3: Our FLEXTPP-C framework allows structured prediction tasks such as annotating events in an input time series with full uncertainty quantification. (a) Example input time series from our synthetic dataset. (b) Ground truth events including per-event-type properties, one row per event type, see legend on top right for abbreviations. (c) Predicting the event type $p(m^{\text{type}}|t, C)$ at a given time $t$ cannot properly describe several events happening at the same time, it instead expresses uncertainty in the event class. (d) A discrete-mark MTPP only predicts start times and event types. (e) Our model samples meaningful sequences as detailed as the ground truth. Error bars are the standard deviation from 256 model samples.

Table 4: We formulate probabilistic time series annotation as a conditional MTPP. Our FLEXTPP-C achieves the best AUC ROC scores by jointly sampling events, event types and durations. FLEXTPP without condition naturally achieves random AUC ROC (0.5), but yields a useful baseline for negative log-likelihoods (NLL). A conditional MTPP with only discrete marks does not predict event durations, but achieves decent AUC ROC when assuming the average duration for each event. As an additional baseline, we compare to a CNN-based time series classifier [Zhao et al., 2017] with limited window size. Standard deviations estimated from five runs, note that the type NLL is conditioned on correct event duration. Time + Type NLL in bits/event, Full NLL in bits/dim.

| Model | NLL (↓) | | AUC ROC (↑) | | | | | |
| | Time + Type NLL | Full NLL | External force | Damping change | Base frequency | Position noise | No Event | Mean Single-class |
| --- | --- | --- | --- | --- | --- | --- | --- | --- |
| FLEXTPP | 0.250 (0.005) | -0.010 (0.003) | 0.5 (0.0) | 0.5 (0.0) | 0.5 (0.0) | 0.5 (0.0) | 0.5 (0.0) | 0.5 (0.0) |
| Only Discrete Marks | -0.474 (0.006) | ✗ | ✗ | ✗ | ✗ | ✗ | ✗ | ✗ |
| + avg duration | ✗ | ✗ | 0.96 (0.01) | 0.63 (0.02) | 0.990 (0.003) | 0.98 (0.01) | 0.91 (0.01) | 0.90 (0.01) |
| FLEXTPP-C | **-0.867** (0.005) | **-0.930** (0.007) | **0.990** (0.01) | **0.66** (0.02) | **0.998** (0.001) | **0.997** (0.003) | **0.93** (0.01) | **0.92** (0.01) |
| Sliding Window Classifier | ✗ | ✗ | 0.86 (0.06) | 0.50 (0.01) | 0.96 (0.04) | 0.87 (0.06) | 0.89 (0.04) | 0.82 (0.02) |

window size. We evaluate all methods using the area under the receiver-operator curve (AUC ROC) [Hanley and McNeil, 1982], a common metric to identify how well predicted events overlap with the ground truth [Schmidl et al., 2022]. FLEXTPP-C annotates most event types almost perfectly, and outperforms the random baseline significantly on the difficult "damping change" event type, which is hard to detect by construction. Naively, discrete mark MTPPs cannot be evaluated with AUC ROC since they do not predict event durations, so we evaluate this metric using the average event duration.

## 5.4 Extracting Event Dependencies

How do events depend on one another under a learned model? Some neural Hawkes process variants provide interpretability through structures their explicit triggering kernels $q(t_i, m_i, t, m)$, so that $\lambda(t_i, m_i|\mathcal{H}_{t_i}) = \sum_{(t,m) \in \mathcal{H}_{t_i}} q(t_i, m_i, t, m)$ [Isik et al., 2023, Zhu et al., 2022]. Such kernels directly encode pairwise influence structures, providing global access to the modeled dependency structure.

Our framework offers a complementary, local interpretability mechanism. To derive it analogously, we can solve eq. (2) for the intensity function $\lambda(t_i|\mathcal{H}_{t_i})$ given a time density $p(t_i|\mathcal{H}_{t_i})$:

$$\lambda(t_i, m_i|\mathcal{H}_{t_i}) = \frac{p(t_i, m_i|\mathcal{H}_{t_i})}{1 - \int_{t_{i-1}}^{t} p(s, m_i|\mathcal{H}_{t_i})ds} \tag{14}$$

Here, we have reintroduced marks $m_i$.

We can extract the local triggering kernel via eq. (14) by evaluating $\lambda(t_2, m_2 \mid (t_1, m_1))$. To demonstrate this, fig. 7 in appendix A.4 replicates a synthetic setup from Isik et al. [2023]. We show

that our model can recover an influence curve matching the ground truth with RMSE 0.01, indicating that our learned event dependencies are both accurate and recoverable.

These approaches exhibit a broader tradeoff between global interpretability and modeling flexibility. Kernel-based models provide a concise, fully inspectable description of event interactions, but their rigidity can misrepresent complex real-world dynamics. For instance, some domains exhibit state-dependent effects—such as advertising "fatigue," where repeated exposures first increase and later decrease response probability—that cannot be captured well by monotone additive influence kernels. In such cases, the model is not expressive enough for the true dependency structure, and the resulting explanations appear structured but do not reflect the true data-generating process.

Our model instead supports local interpretability: one can ask how a specific event or subset of events affects intensities or subsequent event sequences, yielding contextual, fine-grained explanations. This aligns with complex event sequences, where influences are often situational rather than universal.

## 6 Conclusion

In this work, we propose a simple yet powerful method to model more general Marked Time Point Processes — ones with mixed-type (discrete and continuous) marks and auxiliary contextual information. Our intensity-free modeling approach treats MTPP as one joint sequence consisting of auxiliary information (if available), event arrival times, and marks; we then model the sequence autoregressively using a single Transformer with appropriate output heads. We observe that incorporating additional information generally improves modeling performance; this is expected from an information-theoretic perspective — conditioning reduces entropy, which is the minimum achievable negative log-likelihood [Cover and Thomas, 2006]. From a modeling perspective, we observe that complex intensity functions may not be necessary — intensity-free modeling with good one-dimensional density estimators may be enough to model temporal processes well.

Our generalized MTPP framework also highlights how MTPPs can be used as a versatile prediction tool for annotating input time series. Crucially, MTPPs naturally model the joint probability of each prediction, so that each sampled annotation from the model is a consistent explanation of the input. The formulation also naturally allows for overlapping events. We envision future work to extend this paradigm for annotating spatio-temporal sequences to enhance uncertainty quantification in the annotation of time series, such as audio, video, as well as scientific measurements.

**Broader Impact.** Modeling critical data, such as in the context of medicine, can cause harm through wrong or wrongly interpreted predictions, such as those arising from biases in the training data and distribution shifts. On the positive side, modeling additional variables and incorporating context can increase prediction accuracy and enable novel applications.

## 7 Limitations

**Limitations of autoregressive modeling.** Our approach inherits the limitations of autoregressive modeling. In particular, when scaling to high-dimensional marks such as images, alternative generative modeling approaches may be more suitable [Chang et al., 2025]. Another direction is to jointly learn a representation of the marks that is better suited for downstream modeling [Tschannen et al., 2024]. Similarly, if the marks follow special structure such as special geometry or topology, autoregressive models cannot be applied faithfully and generic methods such as [Sorrenson et al., 2024] can be used to model these dimensions.

**Alternative data types.** In our work, we consider discrete and continuous data. Other mark modalities, such as sets or ordered discrete variables, could be studied in the future.

**Limitations of Transformer backbone.** The compute of Transformers scale quadratically with the length $L$ of the underlying sequence: $O(L^2)$. This complexity can make them unsuitable for modeling very long sequences. However by construction, our framework is compatible with other autoregressive models such as recurrent neural networks [Elman, 1990], linear-attention Transformers [Katharopoulos et al., 2020], or state-space models [Gu et al., 2021] that have better length scaling.

## Acknowledgments and Disclosure of Funding

We thank Kushagra Pandey for additional discussions and feedback. This project was funded through support from the Chan Zuckerberg Initiative. Additionally, Stephan Mandt acknowledges funding from the National Science Foundation (NSF) through an NSF CAREER Award IIS-2047418, IIS-2007719, the NSF LEAP Center, and the Hasso Plattner Research Center at UCI. Lukas Laskowski acknowledges financial support from SAP SE. Parts of this research were supported by the Intelligence Advanced Research Projects Activity (IARPA) via Department of Interior/ Interior Business Center (DOI/IBC) contract number 140D0423C0075. The U.S. Government is authorized to reproduce and distribute reprints for Governmental purposes notwithstanding any copyright annotation thereon. Disclaimer: The views and conclusions contained herein are those of the authors and should not be interpreted as necessarily representing the official policies or endorsements, either expressed or implied, of IARPA, DOI/IBC, or the U.S. Government.

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

# Appendix:
# Transformers for Mixed-type Event Sequences

# A    Experimental Details

We base our code on PyTorch [Paszke et al., 2019], PyTorch Lightning [Falcon and The PyTorch Lightning team, 2019], numpy [Harris et al., 2020], hydra [Yadan, 2019] and pandas [The pandas development team, 2020, McKinney, 2010]. Our rational-quadratic spline [Durkan et al., 2019] implementation is adapted from FrEIA [Ardizzone et al., 2018].

Unless stated otherwise, we use the AdamW optimizer [Loshchilov and Hutter, 2019] and a one cycle learning rate scheduler [Smith and Topin, 2019] in the PyTorch implementation with a `div_factor` of 10 and a `pct_start` of 0.1.

## A.1    EasyTPP

The EasyTPP [Xue et al., 2024] benchmark contains the following datasets:

**Amazon** [Ni et al., 2019] features user reviews where each product category serves as a distinct mark. **Retweet** [Zhao et al., 2015] forecasts the popularity of retweet cascades, with event types categorized by user influence levels based on follower counts. **Taxi** [Whong, 2014] captures pickup and dropoff events across New York City, where marks are constructed from the Cartesian product of five predefined locations and two action types. **Taobao** [Xue et al., 2022] models user browsing behavior on an e-commerce platform, using item categories as marks. Lastly, **StackOverflow** collects user badges over time [Du et al., 2016].

Table 5 shows the hyperparameters we use. We identified them using a grid search optimizing for validation negative log-likelihood, separately for each dataset. This mirrors the procedure from which we obtain our baselines [Chang et al., 2025]. We stop training early when the validation NLL has not improved over 300 epochs, and use the checkpoint of best validation for evaluating the model.

We ran all experiments in parallel on a single NVIDIA H100 GPUs, taking about 3 hours for all experiments to finish.

Table 5: Hyperparameters used for the EasyTPP benchmark.

|  | Amazon | Retweet | Taobao | Taxi | StackOverflow |
|---|---|---|---|---|---|
| $n_{\text{epochs}}$ |  |  | 1000 |  |  |
| $n_{\text{head}}$ |  |  | 4 |  |  |
| $n_{\text{ff}}$ |  |  | 256 |  |  |
| Non-linearity |  |  | GELU |  |  |
| Transformer depth | 5 | 5 | 4 | 5 | 5 |
| $d_K$ | 12 | 32 | 12 | 24 | 12 |
| $p_{\text{dropout}}$ | 0.4 | 0.2 | 0.38 | 0.44 | 0.46 |
| $n_{\text{bins}}$ | 11 | 10 | 29 | 8 | 10 |
| Batch size | 159 | 246 | 207 | 176 | 204 |
| Learning rate | 0.001 | 0.00015 | 0.0008 | 0.0005 | 0.0004 |

Table 6: Our method outperforms the previous intensity-free IFTPP [Shchur et al., 2020] on all EasyTPP datasets [Xue et al., 2024] in terms of negative log-likelihood (lower is better, lowest value in **bold**, second lowest value underlined). *(Top)* Extract of table 2 comparing our FLEXTPP to IFTPP. *(Bottom)* Ablation of our transformer backbone with IFTPP's prediction head for continuous data, a mixture of $K$ log-normal distributions.

| Model | Amazon | Retweet | Taxi | Taobao | StackOverflow |
|---|---|---|---|---|---|
| FLEXTPP (Ours) | **-0.633** (0.039) | **5.646** (0.070) | **-0.763** (0.005) | **-1.402** (0.013) | **2.133** (0.004) |
| IFTPP [Shchur et al., 2020] | -0.496 (0.002) | 10.344 (0.016) | -0.453 (0.002) | -1.318 (0.017) | 2.233 (0.009) |
| $K = 16$ log-normal | 2.211 (0.004) | 6.450 (0.001) | -0.734 (0.002) | -1.354 (0.004) | 2.253 (0.005) |
| $K = 32$ log-normal | 2.210 (0.018) | 6.449 (0.001) | -0.732 (0.004) | -1.354 (0.006) | 2.251 (0.004) |
| $K = 64$ log-normal | 2.208 (0.007) | 6.450 (0.001) | -0.727 (0.004) | -1.355 (0.004) | 2.263 (0.006) |
| $K = 128$ log-normal | 2.215 (0.003) | 6.451 (0.001) | -0.719 (0.006) | -1.336 (0.003) | 2.266 (0.003) |

Table 7: Hyperparameters used for EHRSHOT.

| $n_{epochs}$ | $n_{head}$ | $n_{ff}$ | Non-linearity | Transformer depth | $d_K$ | $n_{bins}$ | Batch size | Learning rate |
|---|---|---|---|---|---|---|---|---|
| 2000 | 4 | 256 | GELU | 2 | 16 | 16 | 128 | 0.0005 |

## A.2 Patient Record Data

**EHRSHOT** [Wornow et al., 2023] models irregular time series of clinical events from electronic health records, using lab tests and procedures as distinct marks.

The dataset has been split into Train-Validation-Test with a ratio of 70%-15%-15%. We pick the top 20 most common diseases as subdatasets to train and test the models. For each disease, we capture the top 10 most frequent Current Procedural Terminology (CPT-4) codes that denote medical services and procedures as discrete events, and model the 5 most relevant disease-related lab results as events with their type and continuous value.

We perform early stopping during training when the validation NLL has stopped decreasing for over 200 epochs, and use the checkpoint of best of validation NLL for testing.

To make a fair comparison, all the model share the same model structure. Sinusoidal positional encoding are applied across all models, within each event. The positions are encoded within each event. For each event, arrival time, event type (lab test or procedure), lab test/procedure type, and lab test result (if applicable) are placed at positions 1, 2, 3, and 4, respectively. The metadata (age, gender, ethnicity, and race) are fed as additional tokens into the beginning of the time series, serving as the condition part of the model. More details of the model structure can be found in table 7.

For hyperparameter tuning, the best hyperparameters are chosen based on the validation set. The best batch size is 128 and the best learning rate is 0.0005, searched from the spaces [64, 128, 256] and [0.0001, 0.0005, 0.001] across all models and subdatasets. The best dropout rates are listed in table 8, searched from the space [0.0, 0.1, 0.2].

We ran all experiments in parallel on a single NVIDIA Quadro RTX 8000 GPU and a single NVIDIA Quadro RTX 5000 GPU, taking about 120 hours for all experiments to finish.

## A.3 Conditional Event Detection

### A.3.1 Dataset

We generate our dataset by integrating the SDE described in eq. (13). We give the default values for all parameters in table 11; some values are fixed for all time series and some are sampled anew for each time series.

These parameters are then modified during the SDE integration. Recovering these modifications will later be the prediction to be made by the model.

To this end, we first sample the number of perturbation events to occur (compare table 11). Then, we sample the start time, duration and a ramp time for each modification event according to table 12.

Table 8: Dropout Rates used for EHRSHOT.

| Dataset | Procedures | + Lab Tests | ESGPT | ESGPT-C | FLEXTPP | FLEXTPP-C |
|---|---|---|---|---|---|---|
| Type 2 diabetes mellitus | 0.0 | 0.0 | 0.0 | 0.0 | 0.0 | 0.0 |
| Transplanted kidney | 0.0 | 0.0 | 0.1 | 0.1 | 0.1 | 0.1 |
| Transplanted lung | 0.1 | 0.0 | 0.1 | 0.1 | 0.1 | 0.1 |
| Dyspnea | 0.0 | 0.0 | 0.0 | 0.0 | 0.0 | 0.0 |
| Atrial fibrillation | 0.0 | 0.1 | 0.0 | 0.0 | 0.0 | 0.0 |
| Cardiac transplant disorder | 0.0 | 0.0 | 0.1 | 0.1 | 0.1 | 0.1 |
| End-stage renal disease | 0.1 | 0.0 | 0.1 | 0.1 | 0.1 | 0.1 |
| Transplanted heart | 0.1 | 0.0 | 0.0 | 0.0 | 0.0 | 0.0 |
| Congestive heart failure | 0.0 | 0.1 | 0.0 | 0.0 | 0.0 | 0.0 |
| Chronic pain | 0.0 | 0.1 | 0.0 | 0.0 | 0.0 | 0.0 |
| Neoplasm of female breast | 0.1 | 0.0 | 0.1 | 0.1 | 0.1 | 0.1 |
| Obstructive sleep apnea | 0.1 | 0.0 | 0.0 | 0.0 | 0.0 | 0.0 |
| Diabetes with complication | 0.0 | 0.0 | 0.1 | 0.1 | 0.1 | 0.1 |
| Anemia | 0.0 | 0.0 | 0.0 | 0.0 | 0.0 | 0.0 |
| Coronary artery disease | 0.0 | 0.0 | 0.0 | 0.0 | 0.0 | 0.0 |
| Hypothyroidism | 0.0 | 0.0 | 0.1 | 0.1 | 0.1 | 0.1 |
| Acute myeloid leukemia | 0.1 | 0.0 | 0.1 | 0.1 | 0.1 | 0.1 |
| Depressive disorder | 0.0 | 0.0 | 0.0 | 0.0 | 0.0 | 0.0 |
| Transplanted liver | 0.0 | 0.0 | 0.1 | 0.1 | 0.1 | 0.1 |
| Acute kidney injury | 0.0 | 0.1 | 0.0 | 0.0 | 0.0 | 0.0 |

Table 9: Table 3 **Time + Discrete Procedure NLL** ($\downarrow$) with standard deviations over five runs.

| Dataset | Procedures | + Lab Tests | ESGPT | ESGPT-C | FLEXTPP | FLEXTPP-C |
|---|---|---|---|---|---|---|
| Type 2 diabetes mellitus | 0.691 (0.0020) | 0.202 (0.0012) | 0.186 (0.0010) | 0.182 (0.0010) | 0.147 (0.0006) | **0.143** (0.0012) |
| Transplanted kidney | 0.755 (0.0130) | 0.170 (0.0021) | 0.199 (0.0020) | 0.192 (0.0028) | 0.134 (0.0007) | **0.130** (0.0014) |
| Transplanted lung | 0.912 (0.0019) | 0.089 (0.0003) | 0.078 (0.0006) | 0.075 (0.0002) | 0.068 (0.0003) | **0.063** (0.0000) |
| Dyspnea | 0.681 (0.0019) | 0.162 (0.0003) | 0.162 (0.0011) | 0.155 (0.0011) | 0.121 (0.0006) | **0.117** (0.0005) |
| Atrial fibrillation | 0.715 (0.0072) | 0.081 (0.0001) | 0.075 (0.0003) | 0.071 (0.0003) | 0.062 (0.0001) | **0.059** (0.0005) |
| Cardiac transplant disorder | 0.887 (0.0055) | 0.034 (0.0001) | 0.034 (0.0004) | 0.033 (0.0005) | 0.029 (0.0002) | **0.027** (0.0004) |
| End-stage renal disease | 0.697 (0.0035) | 0.166 (0.0012) | 0.169 (0.0020) | 0.164 (0.0018) | 0.128 (0.0012) | **0.117** (0.0008) |
| Transplanted heart | 0.034 (0.0024) | 0.887 (0.0001) | 0.033 (0.0002) | 0.032 (0.0004) | 0.029 (0.0003) | **0.028** (0.0001) |
| Congestive heart failure | 0.134 (0.0040) | 0.704 (0.0006) | 0.130 (0.0012) | 0.128 (0.0026) | 0.103 (0.0007) | **0.100** (0.0005) |
| Chronic pain | 0.735 (0.0042) | 0.099 (0.0002) | 0.095 (0.0004) | 0.093 (0.0006) | 0.076 (0.0017) | **0.073** (0.0009) |
| Neoplasm of female breast | 0.716 (0.0025) | 0.043 (0.0007) | 0.040 (0.0006) | 0.038 (0.0003) | 0.032 (0.0003) | **0.030** (0.0006) |
| Obstructive sleep apnea | 0.695 (0.0023) | 0.093 (0.0006) | 0.090 (0.0007) | 0.089 (0.0013) | 0.061 (0.0006) | **0.059** (0.0002) |
| Diabetes with complication | 0.718 (0.0022) | 0.176 (0.0004) | 0.164 (0.0017) | 0.160 (0.0013) | 0.133 (0.0007) | **0.129** (0.0005) |
| Anemia | 0.655 (0.0025) | 0.164 (0.0013) | 0.165 (0.0012) | 0.158 (0.0007) | 0.127 (0.0006) | **0.126** (0.0008) |
| Coronary artery disease | 0.692 (0.0021) | 0.144 (0.0011) | 0.138 (0.0008) | 0.134 (0.0012) | 0.108 (0.0006) | **0.105** (0.0003) |
| Hypothyroidism | 0.735 (0.0013) | 0.224 (0.0010) | 0.212 (0.0011) | 0.208 (0.0013) | 0.166 (0.0007) | **0.159** (0.0009) |
| Acute myeloid leukemia | 0.729 (0.0065) | 0.182 (0.0021) | 0.160 (0.0011) | 0.154 (0.0012) | 0.124 (0.0009) | **0.117** (0.0005) |
| Depressive disorder | 0.683 (0.0027) | 0.136 (0.0008) | 0.135 (0.0011) | 0.129 (0.0007) | 0.108 (0.0006) | **0.105** (0.0013) |
| Transplanted liver | 0.799 (0.0063) | 0.183 (0.0008) | 0.192 (0.0015) | 0.186 (0.0024) | 0.144 (0.0019) | **0.142** (0.0026) |
| Acute kidney injury | 0.201 (0.0026) | 0.645 (0.0003) | 0.191 (0.0006) | 0.185 (0.0004) | 0.144 (0.0010) | **0.139** (0.0005) |

Depending on the type, we sample the event properties via table 13. The ramp time avoids sudden changes in parameters, so that the events are harder to detect.

The events replace the parameters in eq. (13) with the following dynamic coefficients:

$$\sigma_x(t) = \sigma_{x,0} + \sum_{E:\, m_E^{\text{type}}=D} \alpha(t; E)\sigma_{x,E}, \tag{15}$$

$$\omega(t) = w_0 \prod_{E:\, m_E^{\text{type}}=C} (1 + \alpha(t; E)\omega_E), \tag{16}$$

$$\gamma(t) = \gamma_0 \prod_{E:\, m_E^{\text{type}}=B} (1 + \alpha(t; E)\gamma_E), \tag{17}$$

$$f(t) = \sum_{E:\, m_E^{\text{type}}=A} \alpha(t; E)\beta s_E \left( t - \frac{2\pi}{T} t \right). \tag{18}$$

Here, the sums/products run over all events $E$ of the corresponding type $m_E^{\text{type}}$. The function $\alpha(t; E)$ turns off the event outside of its domain and interpolates over the ramp time at the beginning and end

Table 10: Table 3 **Full NLL** (↓) with standard deviations over five runs.

| Dataset | ESGPT | ESGPT-C | FLEXTPP | FLEXTPP-C |
|---|---|---|---|---|
| Type 2 diabetes mellitus | 0.754 (0.0038) | 0.758 (0.0056) | 0.641 (0.0050) | **0.638** (0.0027) |
| Transplanted kidney | 0.859 (0.0070) | 0.863 (0.0225) | 0.670 (0.0083) | **0.668** (0.0111) |
| Transplanted lung | 0.851 (0.0013) | 0.857 (0.0200) | 0.581 (0.0060) | **0.546** (0.0122) |
| Dyspnea | 0.560 (0.0111) | 0.549 (0.0185) | 0.463 (0.0038) | **0.438** (0.0021) |
| Atrial fibrillation | 0.388 (0.0119) | 0.376 (0.0091) | 0.204 (0.0048) | **0.192** (0.0090) |
| Cardiac transplant disorder | 0.552 (0.0524) | 0.594 (0.0553) | 0.408 (0.0116) | **0.393** (0.0203) |
| End-stage renal disease | 0.757 (0.0145) | 0.753 (0.0112) | 0.580 (0.0037) | **0.578** (0.0032) |
| Transplanted heart | 0.736 (0.0233) | 0.701 (0.0163) | 0.546 (0.0041) | **0.521** (0.0036) |
| Congestive heart failure | 0.846 (0.0102) | 0.861 (0.0116) | 0.525 (0.0066) | **0.509** (0.0096) |
| Chronic pain | 0.765 (0.0046) | 0.743 (0.0075) | **0.162** (0.1045) | 0.165 (0.0078) |
| Neoplasm of female breast | 0.608 (0.0068) | 0.608 (0.0141) | 0.516 (0.0309) | **0.461** (0.0270) |
| Obstructive sleep apnea | 0.710 (0.0075) | 0.676 (0.0097) | 0.195 (0.0354) | **0.187** (0.0285) |
| Diabetes with complication | 0.694 (0.0113) | 0.703 (0.0114) | **0.579** (0.0023) | 0.582 (0.0047) |
| Anemia | 0.322 (0.0014) | 0.296 (0.0190) | **0.238** (0.0121) | 0.245 (0.0097) |
| Coronary artery disease | 0.966 (0.0246) | 0.986 (0.0247) | 0.636 (0.0138) | **0.625** (0.0134) |
| Hypothyroidism | 0.666 (0.0068) | 0.664 (0.0063) | **0.524** (0.0037) | 0.528 (0.0080) |
| Acute myeloid leukemia | 0.727 (0.0131) | 0.735 (0.0139) | 0.567 (0.0161) | **0.561** (0.0050) |
| Depressive disorder | 0.813 (0.0038) | 0.826 (0.0101) | 0.608 (0.0129) | **0.545** (0.0133) |
| Transplanted liver | 0.907 (0.0211) | 0.902 (0.0150) | 0.727 (0.0104) | **0.725** (0.0159) |
| Acute kidney injury | 0.706 (0.0037) | 0.711 (0.0037) | 0.621 (0.0042) | **0.605** (0.0028) |

Table 11: Base parameters of the harmonic oscillator specified in eq. (13). The observation noise is randomly sampled for each time series. The notation $\sim \cdot$ indicates that the corresponding value is sampled.

| Property | Value |
|---|---|
| Damping $\gamma_0$ | 0.1 |
| Base frequency $\omega_0$ | $30 \cdot 2\pi$ (30 Hz oscillator) |
| Time range | 0..1 in $10^4$ steps |
| Noise strength on position $\sigma_{x,0}$ | 0.01 |
| Noise strength velocity $\sigma_v$ | 1.0 |
| Observation noise $\epsilon$ | $\sim \mathrm{LogNorm}(\mu = \log(0.01), \sigma = \log(.1))$ |
| Number of perturbation events | $\sim \mathcal{U}(\{1, \ldots, 5\})$ |

Table 12: Shared properties for all events are randomly sampled for each event, independent of other events. The last column indicates whether that property is to be predicted by our models.

| Property | Distribution | To be predicted? |
|---|---|---|
| Event duration $\delta t$ | $\sim \mathcal{U}([.025, 0.5])$ | ✓ |
| Event start $t$ | $\sim \mathcal{U}([0, 1 - \text{duration}])$ | ✓ |
| Event type $m^{\text{type}}$ | $\sim \mathcal{U}(\{\text{types defined in table 13}\})$ | ✓ |
| Ramp time $t_r$ | $\sim \mathcal{U}([0, \text{duration}/4])$ | ✗ |

Table 13: Custom event properties per event type and how we draw them. They are all predicted by our model.

| Event Type | Property | Distribution |
|---|---|---|
| A: External force | Magnitude $\beta$ | $\sim \mathcal{U}([.5, 1.5])$ |
| | Force shape $s$ | $\sim \mathcal{U}(\{\sin, \text{square}, \text{sawtooth}\})$ |
| | Periodicity $T$ | $\sim \mathcal{U}([0.01, 0.1])$ |
| B: Damping change | Factor $\kappa_\omega$ | $\sim \mathcal{U}([0.5, 1.5])$ |
| C: Base frequency change | Factor $\kappa_\gamma$ | $1 + sa$, where $s \sim \mathcal{U}(\{-1, 1\})$ and $a \sim \mathcal{U}([0.1, 0.2])$ |
| D: Position noise | Additional noise $\sigma_x$ | $\sim \mathcal{U}([1.0, 5.0])$ |

of the event:

$$\alpha(t; E) = \begin{cases} 0 & \text{if } t \le t_E \\ \frac{t - t_E}{t_{r,E}} & \text{if } t_E < t < t_E + \delta t_E \\ 1 & \text{if } t_E + t_{r,E} \le t \le t_E + \delta t_E - t_{r,E} \\ 1 - \frac{t - (t_E + \delta t_E - t_{r,E})}{t_{r,E}} & \text{if } t_E < t < t_E + \delta t_E \\ 0 & \text{if } t_E + \delta t_E \le t. \end{cases} \tag{19}$$

We use sdeint's `itoint` [Jaekel, 2015] to solve the SDE, initializing to $x = 1.0, v = 0.0$. After integrating, we add independent observation noise $\epsilon$ to each time point. We avoid extreme values by softly clamping the time series in $[-10, 10]$, where $C = L/2$ and $L = 10$:

$$\text{soft\_clip}(x) = \text{sign}(x) \cdot \text{clipped} \tag{20}$$

$$\text{where} \quad \text{clipped} = \begin{cases} |x| & \text{if } |x| \le L \\ L + (C - L)\left(1 - e^{-k(|x|-L)}\right) & \text{if } |x| > L \end{cases} \tag{21}$$

$$k = \frac{1}{C - L}. \tag{22}$$

We generate 100,000 input time series with annotations, using 80% for training, and 10% for validation and testing each. We provide the code for generating the data in the supplementary files.

### A.3.2   MTPP-based event detection

It is straightforward to turn the prediction of the above events into a generative prediction task via an MTPP.

To this end, we utilize an event $E$'s start time $t_E$ as its arrival time, the type of change per table 13 as the discrete mark type $m_E^{\text{type}}$ and the duration as well as the other properties as the additional marks. Concretely:

$$N(A) = 4, \qquad\qquad d(A) = (\text{cont}, \text{cont}, \text{disc}, \text{cont}), \tag{23}$$
$$N(B) = N(C) = N(D) = 2, \qquad d(B) = d(C) = d(D) = (\text{cont}, \text{cont}). \tag{24}$$

As a conditioning network, we preprocess the input time series using a Transformer with full attention (non-causal). It shares the embedding dimension $d_E = n_{\text{head}} d_K$ with the autoregressive backbone of the MTPP model. To tokenize the data, we split the incoming time series into windows of size $l$, and map them through a linear layer to the embedding size $d_E$. The conditioning Transformer has depth 2, $d_{\text{ff}} = 256$ and dropout 0.1.

All other hyperparameters are specified in Table 14, each determined from a hyperparameter optimization. Interestingly, the discrete-only variant has significantly more monotonic bins in the continuous distribution than our FLEXTPP-C. We think that this is due to the requirement to model a more complicated distribution for the next-time prediction. We stop training if the last 300 epochs did not reduce the validation loss and evaluate each run using the checkpoint with minimal validation loss. Training takes about three hours per run, with five parallel runs on a single NVIDIA H100.

We make use of two complementary positional encodings: A learned embedding encoding the position in the event, and a positional encoding of the event in the sequence.

### A.3.3   Ablation on Event Property Order

In an autoregressive model, we can change the order of variables to obtain the same joint distribution via the chain rule of probability:

$$p_D(X_1, \dots X_L) = \prod_{l=1}^{L} p_D(X_l | X_1, \dots, X_{l-1}) = \prod_{l=1}^{L} p_D(X_{\pi(l)} | X_{\pi(1)}, \dots, X_{\pi(l-1)}). \tag{25}$$

We find that for the event annotation dataset, it is beneficial to deviate from the order in eq. (3):

$$(t_i, m_i, m_i^{\text{add}}) = (t_i, m_i, \text{duration}_i, \text{other event properties}_i). \tag{26}$$

Table 14: Hyperparameters of the MTPP-based model.

| | Discrete mark MTPP | FLEXTPP-C |
|---|---|---|
| $n_{\text{epochs}}$ | 1000 | |
| $n_{\text{head}}$ | 4 | |
| $n_{\text{ff}}$ | 256 | |
| Non-linearity | GELU | |
| Transformer depth | 7 | |
| $d_K$ | 28 | |
| $p_{\text{dropout}}$ | 0.25 | 0.2 |
| $n_{\text{bins}}$ | 20 | 8 |
| Batch size | 160 | |
| Learning rate | 0.001 | |
| Conditioning window size $l$ | 200 | 240 |

Table 15: Ablation comparing different orderings of each event's data for FLEXTPP-C in the event annotation task. Event parts: **S**tart of event, **T**ype of event, **D**uration, and remaining **P**roperties. Values in brackets are standard deviations over eight runs. Time + Type NLL in bits/event, Full NLL in bits/dim.

| Order | Time + Type NLL ($\downarrow$) | Full NLL ($\downarrow$) |
|---|---|---|
| SDTP | **-0.867** (0.005) | **-0.930** (0.007) |
| STDP | -0.596 (0.004) | -0.778 (0.006) |
| STPD | -0.595 (0.005) | -0.775 (0.006) |

and instead model in the order:

$$(t_i, \text{duration}_i, m_i, \text{other event properties}_i). \tag{27}$$

Table 15 compares the validation negative log-likelihoods of different orders. Note that hyperparameters were optimized for eq. (27), and thus we cannot exclude that there are hyperparameters for which the other orders perform just as well (for example, increase the number of bins of the spline to model more complex distributions in alternative orderings). In the end, eq. (25) suggests that the full likelihoods should be identical across reordering the data.

### A.3.4 Baseline: CNN-based Time Series Classifier

We compare FLEXTPP-C to a CNN-based classifier for conditional event detection. We follow the structure from the time series anomaly detection survey by Schmidl et al. [2022] and make use of convolutional neural networks as classifiers [Zhao et al., 2017]. The approach uses a classifier that predicts the class at a time point $t \in [0, 1]$ of what events it observes given a slice of the time series of size $l$ around $t$. At inference time, this classifier is applied in a sliding-window approach. The finite window size of the classifier limits its capability to detect events. We assume that every timestamp has been annotated by counting the number of events per event type that could occur and write that into a normalized vector $y(t) \in [0, 1]^{m+1}$ to be predicted with $m$ being the number of different event types (plus one type for "no event"). This allows the classifier to model multiple concurrent events through weighted class activations. However, it does not allow it to differentiate this from uncertainty in the prediction, as we analyze in appendix A.3.5.

Following [Schmidl et al., 2022], for training, we cut the input time series into slices of length $l$ with strides $l_s$. We found $l = 500$ and $l_s = 100$ to perform best. Each resulting window is then assigned the label vector of its central timestamp.

Then, we train a convolutional neural network to map the observations to the target vector $y(t)$ using cross-entropy loss. We used Adam optimizer with a learning rate of 0.001 trained for 5 epochs.

Multiple runs were trained concurrently on a single NVIDIA A100 Tensor Core GPU, and each single run completed within 2 hours of starting.

The classifier consists of two convolutional blocks, each followed by batch normalization, a non-linear activation function, temporal downsampling through pooling, and dropout for regularization (see

table 16). The first block increases the number of feature channels, and the second further refines the representation while reducing the temporal resolution. The resulting feature maps are flattened and passed through a fully connected layer with an intermediate activation and additional dropout. A final linear layer maps the representation to the desired number of output classes. To recover per-timestamp predictions from the window-based CNN outputs, we aggregate the predicted label vectors of all windows that include a given timestamp:

$$\hat{y}(t) = \frac{1}{|W(t)|} \sum_{w_i \in W(t)} \hat{y}(w_i), \tag{28}$$

where

$$W(t) := \{w_i \mid t \in w_i\} \tag{29}$$

is the set of all windows that include timestamp $t$. This averaging doubles the field of view for each timestamp's prediction and improves robustness and smoothness over time.

Table 16: Architecture of the 1D CNN used for window-based event classification. The input has shape (batch_size, $1, l + 1$), where $l + 1$ is the window length.

| Layer | Operation |
|---|---|
| Conv1 | Conv1d(1, 32, kernel=3, padding=1) |
| | BatchNorm1d(32) |
| | ReLU() |
| | MaxPool1d(kernel_size=2) |
| | Dropout(p=0.2) |
| Conv2 | Conv1d(32, 64, kernel=3, padding=1) |
| | BatchNorm1d(64) |
| | ReLU() |
| | MaxPool1d(kernel_size=2) |
| | Dropout(p=0.2) |
| Flatten | view(batch_size, -1) |
| FC1 | Linear(64 $\cdot \lfloor \frac{l+1}{4} \rfloor$, 128) |
| | ReLU() |
| | Dropout(p=0.3) |
| FC2 | Linear(128, $m$) |

### A.3.5   Method Comparison

In this section, we expand on the evaluation in section 5.3. We compare the conditional MTPP with only discrete marks + fixed average event length, our FLEXTPP-C also modeling duration and event properties, and a sliding window classifier. We compare via samples, likelihoods, and per-timestamp class probabilities.

**Samples from MTPP approaches**   Figures 4 to 6 shows a total of six randomly selected test time series. First, we show the predictions of the MTPP approaches: Both the classical MTPP (only discrete marks) and our method detect events accurately, except for the damping change, which is hard to detect in the data given its subtle effect. However, the flexible mark structure is more useful since it models event durations and the event properties.

For the MTPP with only discrete marks, we visualize the events with a constant event duration of $(.5 + 0.025)/2 = 0.2625$, which is the average length.

The sliding window classifier does not directly yield such samples since it has no understanding of event starts.

**Likelihoods**   Table 4 compares likelihoods of the MTPP approaches. The resulting classification likelihood from the sliding window classifier is not directly compatible with the likelihoods of our model in eq. (5), since they operate in different spaces.

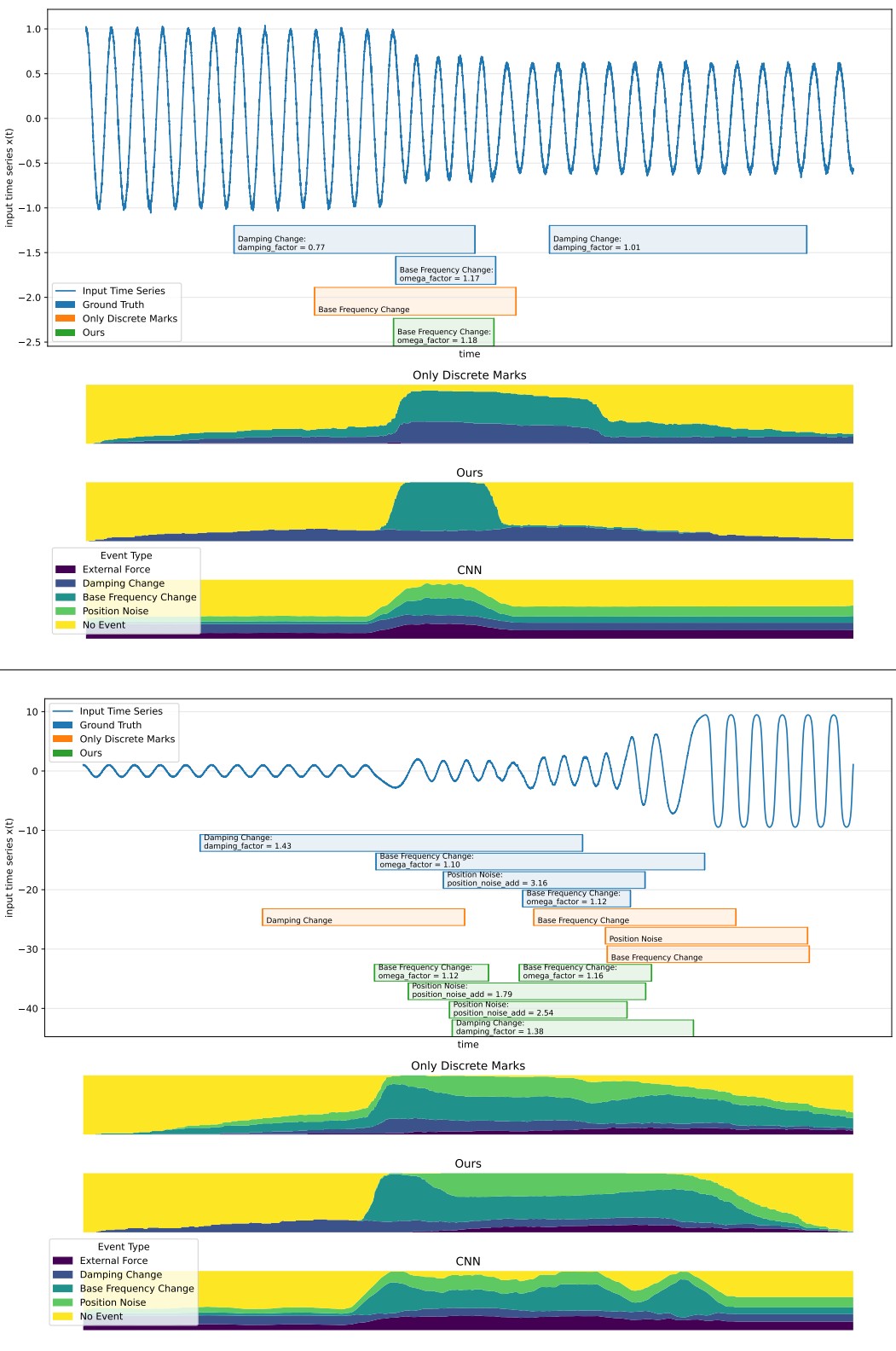

Figure 4: **Uncurated model and baseline predictions on the event prediction task** (1/3). *(In boxes)* Input time series, ground truth event annotations *(blue)*, as well as one random annotation from the discrete-only MTPP model *(orange)* and our flexible-modality model *(green)*. The ground truth and our models also list event properties, which are not available for the discrete-only model. *(Below plot)* Predicted $p(y(t)|$time series) from the three models (ours, discrete-only MTTP, baseline CNN).

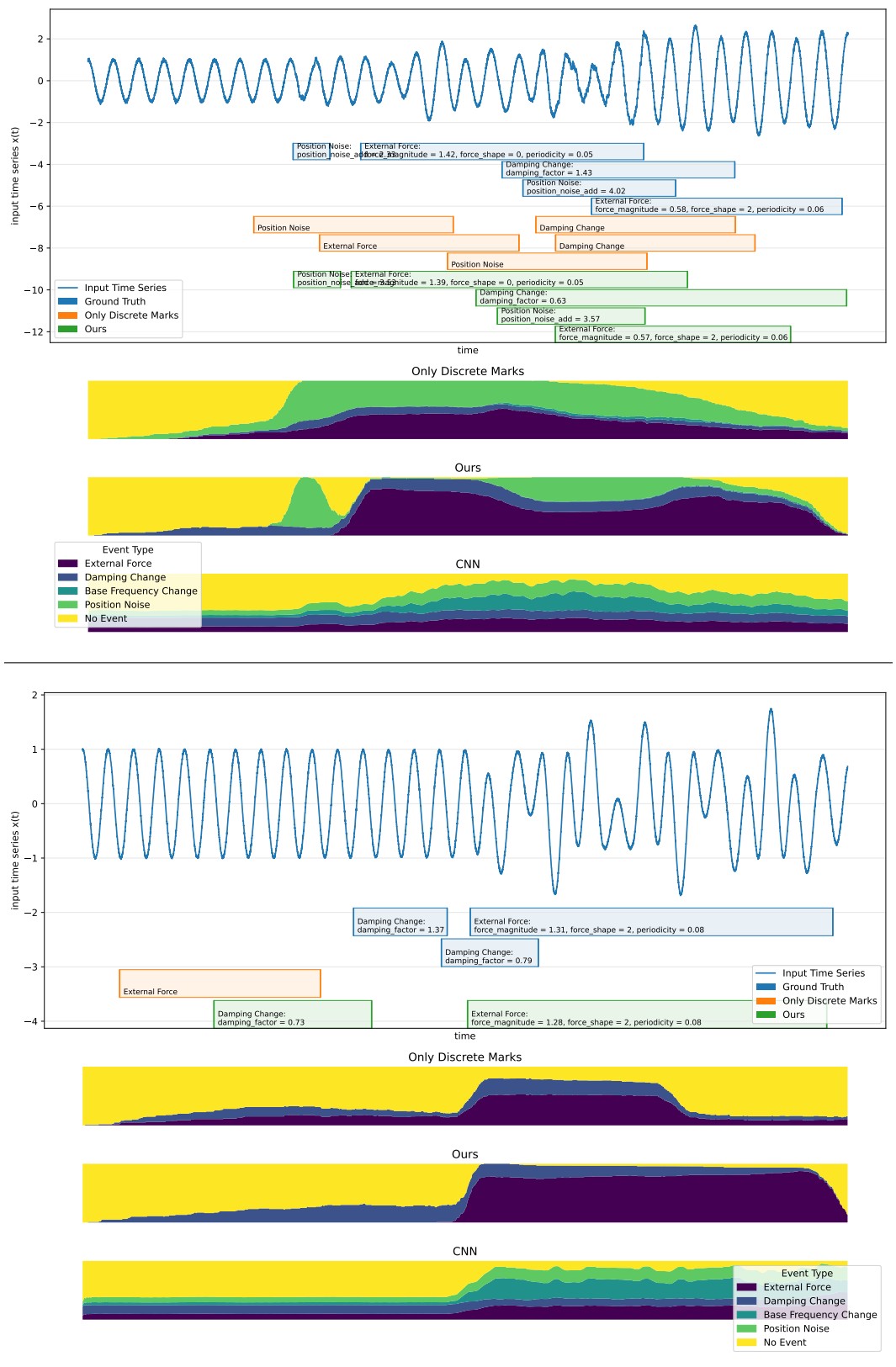

Figure 5: **Uncurated model and baseline predictions on the event prediction task** (2/3). Same as fig. 4.

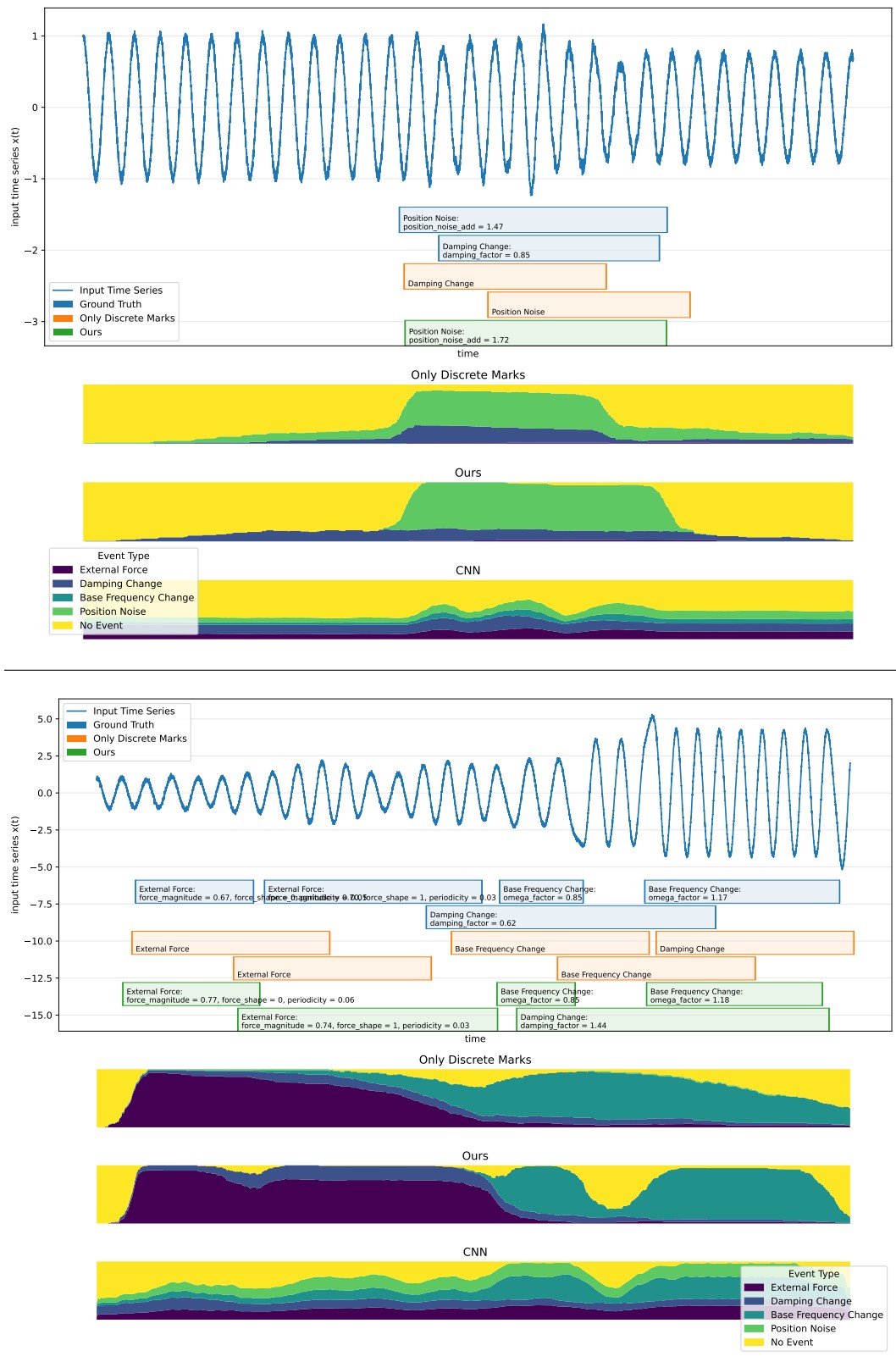

Figure 6: **Uncurated model and baseline predictions on the event prediction task** (3/3). Same as fig. 4.

**Class probabilities per timestamp** To compare the different modeling approaches, we evaluate the class probabilities $p(y(t)|$time series$)$ at each time stamp given the time series. To get these predictions about the class distribution at time $t$ from the MTPP models, we approximate the trained models $p(y(t)|$time series$)$ by averaging event occurrence over 100 sampled event annotations. For the MTPP with only discrete marks, we again use the average event duration for all predicted events.

Figures 4 to 6 visualize the resulting probabilities for the three approaches. One can see that all three methods mostly agree at which time some event starts. However, the discrete-marks MTPP is biased when multiple events occur since it does not predict the duration of the events. Also, it learns heavier distributions for the events.

The classifier is less well calibrated in terms of regards of which class it expects. We think that this is due to a fundamental limitation of the formulation of annotation as classification: By construction, a classifier-based approach cannot differentiate between being unsure about which event class to predict and there being several events present. For example, let's assume there are two event types $A$ and $B$ and the classifier predicts the following values:

$$p(\text{event } A \text{ at } t|\text{window}) = 1/2, \quad p(\text{event } B \text{ at } t|\text{window}) = 1/2, \quad p(\text{no event at } t|\text{window}) = 0. \tag{30}$$

This result could be caused by (a) the model is certain that both events $A$ and $B$ being present, or (b) there is some event, but it is unclear what type of event ($A$ or $B$) there is. We think that this ambiguity in the task representation leads to an overall high uncertainty in the predictions of the classifier.

Table 4 quantify the performance of the models the macro-averaged Receiver Operating Characteristic – Area Under the Curve (ROC AUC) [Hanley and McNeil, 1982] on the per-timestamp $y(t)$ vectors. This metric is computed by adopting a one-vs-rest strategy: for each class, a binary ROC AUC score is calculated by treating the current class as the positive class and all others as negative. The final score is then obtained by averaging the individual AUCs across all classes:

$$\text{Macro AUC} = \frac{1}{m} \sum_{k=1}^{m} \text{AUC}_k. \tag{31}$$

Here, each $AUC_k$ represents the ROC AUC score for class $k = 1, \ldots, m+1$ (compare appendix A.3.4).

Each ROC curve is insensitive to a badly calibrated model because it is based on the ranking of predicted scores, not their absolute values. Therefore, the CNN Classifier still yields useful ROC AUCs despite the notably worse performance in predicting $p(y(t)|$time series$)$ in figs. 4 to 6.

### A.4 Extracting triggering kernel

For this experiment, we sample $N = 100,000$ event sequences from the following intensity function:

$$\lambda(t|\mathcal{H}_t) = \mu + \sum_{\substack{t_i < t \\ t - t_i < \beta}} \alpha \, \sin\left(\frac{\pi(t - t_i)}{\beta}\right). \tag{32}$$

We choose $\mu = 0.1, \alpha = 0.2, \beta = 1$ and sample $T = 4$ events per sequence.

We then train a FLEXTPP with hyperparameters given in table 17.

Figure 7 shows the above kernel and the kernel extracted from our model. To get it from our model, we average over $t_1 = 0.1, 0.2, \ldots, 10.0$ and evaluate the intensity function from the learned probabilities via eq. (14) as a function of $t_2 = t_1 + 0.033, \ldots, t_1 + 10.0$.

## B Details on Architecture

Figure 1 in the main text visualizes our architecture. Figure 8 visualizes the architecture with more details and shows the sampling according to algorithm 1. The multi-head attention block is causal, meaning that token $i$ can only attend to tokens $1, \ldots, i-1$. The transformer works with a token dimension of $d_E = n_{\text{head}} d_K$ and the feed-forward networks in each transformer block have a hidden dimension of $d_{\text{ff}}$.

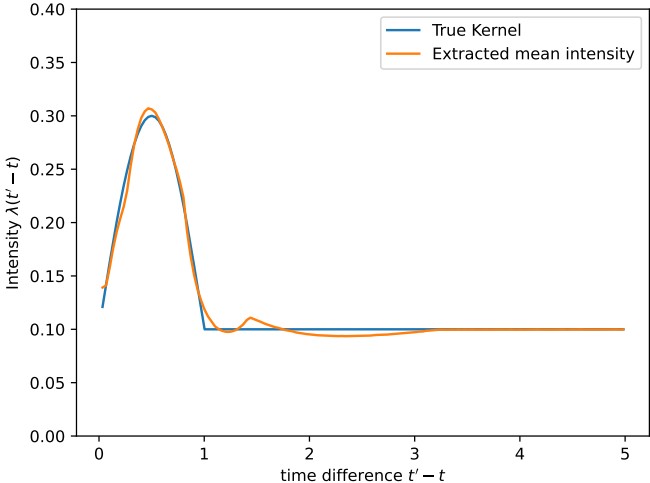

Figure 7: FLEXTPP *(orange)* can accurately learn the triggering kernel in eq. (32) *(blue)* from data.

Table 17: Hyperparameters of the kernel extraction model.

|  | FLEXTPP |
| --- | --- |
| $n_{\text{epochs}}$ | 200 |
| $n_{\text{head}}$ | 4 |
| $n_{\text{ff}}$ | 256 |
| Non-linearity | GELU |
| Transformer depth | 2 |
| $d_K$ | 16 |
| $p_{\text{dropout}}$ | 0.1 |
| $n_{\text{bins}}$ | 10 |
| Batch size | 1024 |
| Learning rate | 0.0008 |

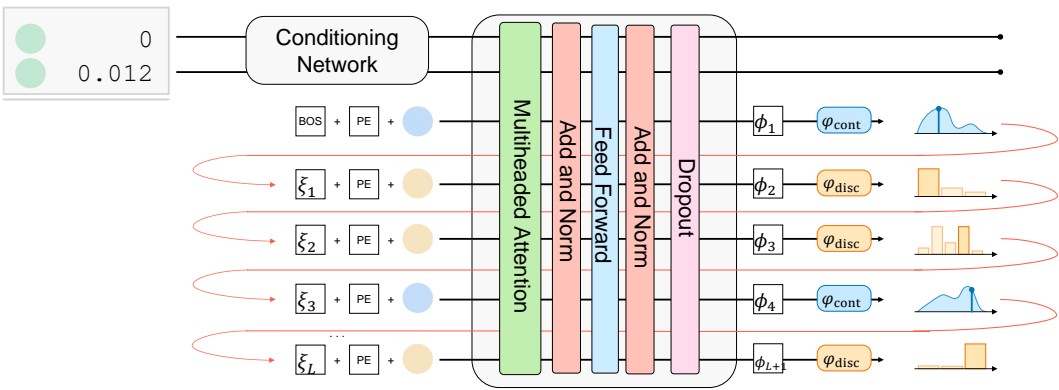

Figure 8: **Details on our transformer architecture and visualization of sampling.** The multi-head attention layer includes both causal attention and dropout. Sampling occurs auto-regressively, where each token is fed into the model after being sampled, until the terminal token is generated.

The output embeddings of the transformers are ignored for the condition tokens, and the others are passed into a small neural network predicting the parameters of the continuous respectively discrete distributions each. These networks $\varphi_{\text{cont}}$ respectively $\varphi_{\text{disc}}$ have one hidden layer with hidden width $2\max\{d_E, d_{\text{cont}}\}$ ($2\max\{d_E, d_{\text{disc}}\}$) a ReLU activation. Even though different positions in the sequence may allow separate numbers of discrete values, we find that we never sample a disallowed class at inference (indicating that the network has learned the sequence accurately) and so we do not need to track the actual number of allowed classes depending on position. Instead, we set the number of classes to the maximum number of classes any token has to support. For values that are by construction positive (that is all arrival times relative to the previous event, as well as durations for the time series annotation experiments), we first take the logarithm of these values. We also reflect this in the likelihood computation, which is based on linear time.

This is how we feed data into the transformer:

First, we preprocess the condition with some network to transform from its corresponding modality to one or several tokens (for example the large vector for the time is not directly compatible with the transformer token-based embedding). We provide the details in each experimental section in appendix A.

Then, we embed each *value* in $X$ as a token in eq. (3) depending on its type: For discrete entries $D_i = \text{disc}$, we use a learned embedding vector for each possible class. For continuous entries, we learn an affine layer $wx + b$, where $w, b \in \mathbb{R}^{d_m}$ and $x \in R$ is the scalar to be embedded. We shift these tokens to the next position, since the $i$th output of the transformer is mapped to the distribution $p(X_i|X_{<i})$, which must only have access to the history. The first entry is replaced with $0$ as a beginning of sequence identification (there is no history for the first entry).

We also embed the *data type* in $D$ ($\in \{\text{disc}, \text{cont}\}$) to be expected for each token using a learned embedding. These embeddings are added to the shifted value tokens. They are *not shifted*, the type of entry can be determined at inference time using algorithm 1.

Finally, we add a positional encoding to inform the model about where it is in the sequence. We give the details on the positional encoding and the conditioning networks for each case separately.

If the number of events per sequence is not constant over the dataset, we mark the end of the sequence with a special "EOS" mark type ($m_{T+1}^{\text{type}} = \text{EOS}$). Note that since the end of sequence is never fed into the model as it is the last entry in a sequence and all values are shifted by one. Instead the "EOS" event only enters when evaluating the last sequence entry's distribution after the transformer.

