# OpenReview forum: "Transformers for Mixed-type Event Sequences"
_NeurIPS.cc/2025/Conference — NeurIPS 2025 spotlight_

### Official Review · Reviewer_ByqH · 2025-06-20

**Clarity:** 2
**Significance:** 3
**Originality:** 1
**Rating:** 5
**Confidence:** 4

**Summary:**

This paper proposes a method for learning mixed-type Marked Temporal Point Processes using normalizing flows and finds that it improves predictive performance on a number of tasks.

**Questions:**

How is this different from https://arxiv.org/abs/2306.11547 and https://arxiv.org/abs/1909.12127? What does this paper add that is not in either of those two papers?

Can you please address the questions about 4.2 listed in weaknesses?

**Ethical Concerns:**

["NO or VERY MINOR ethics concerns only"]

**Final Justification:**

The authors have addressed all of my concerns, in particular the comparisons to prior work.

It's still a bit unclear why this paper got the opposite results of IFTPP (in that this paper found that normalizing flows were helpful while IFTPP found that they were not), and I wish there was more investigation into that, but I'm satisfied with the experimental results as they stand.

**Limitations:**

yes

**Paper Formatting Concerns:**

No concerns

**Quality:**

2

**Strengths And Weaknesses:**

Strengths:
- Marked temporal processes with mixed outcome types are very important

Weaknesses:
- The conditional handling seems overly engineered. Why not just add it as an event at the start of the sequence?
- The lack of public source code makes it very challenging to understand and verify many of the results
- The novelty here is very questionable. Marked Temporal Point Processes with mixed types have been explored before. https://arxiv.org/abs/2306.11547 is a recent example that has a nice open source codebase for handling Marked Temporal Point Process with mixed types.
- Likewise, using normalizing flows for temporal point processes has also been explored. See https://arxiv.org/abs/1909.12127.
- Related to the missing code, many of the experiments require more detailed information. Secton 4.2 is particularly lacking. How many patients are there per condition? How are you addressing the issue of multiple labels per patient inherent in EHR data? In Table 3 when you say "Lab tests w/o values" are you excluding the values only as prediction targets or as both prediction targets and features?
- No notion of uncertainty or statistic soundness of many of the comparisons. This is especially important for small datasets like EHRSHOT.
- Missing ablation experiments to test whether particular components (like the normalizing flows part) are helpful. This is something to remark upon because the current literature (like https://arxiv.org/abs/1909.12127) found that normalizing flows were not helpful.

---

> ### Author Rebuttal · Authors · 2025-07-31
>
> We thank the reviewer for their detailed comments and suggestions, as well as the helpful pointer to the literature:
>
> > What does this paper add that is not in [https://arxiv.org/abs/2306.11547 and https://arxiv.org/abs/1909.12127]
>
> In short: **Our model generalizes and significantly outperforms both approaches.** These are our contributions over the literature:
>
> 1. **Intensity-free outperforms intensity-based**. Previous work argued that intensity-based methods dominate due to "fewer modeling restrictions" [https://arxiv.org/pdf/2412.19634, Section 4]. Our approach brings intensity-free modeling back on the table, outperforming all intensity-based methods (except for DLHP in one dataset, see Table 1), all while avoiding numeric integration at training and rejection sampling at inference time.
> 2. **Point processes beyond homogeneous marks**. To the best of our knowledge, FlexTPP is the first universal point process method for heterogeneous mark spaces. Event Stream GPT (https://arxiv.org/abs/2306.11547) does not use a normalizing flow for continuous marks, so we can model distributions more accurately.
> 3. **Generalizing point processes to new tasks**. The generalized framework allows modeling structured prediction tasks as point processes, which we show with the successful annotation of events in a dense time series. Event Stream GPT is built to predict continuous marks with a Gaussian, which deteriorates sample quality over time and prevents sampling sequences.
> 4. **Successful extraction of triggering kernels**. Based on the suggestion by reviewer M7Vi, we are adding an evaluation on the interpretability of our model to the camera ready version. See our answer to M7Vi for details.
>
>
> Let us now give more details on the two papers.
>
> > Difference to https://arxiv.org/abs/2306.11547
>
> We thank the reviewer for pointing us to Event Stream GPT (ESGPT). The key difference is that **ESGPT models model continuous marks using unimodal Gaussians**, whereas our FlexTPP uses normalizing flows, which can approximate arbitrary distributions.
>
> This flexibility leads to better generative performance: **FlexTPP(-C) achieves lower NLL than ESGPT(-C)** on all twenty but one electronic health dataset. For example, on the first three datasets in Table 3:
>
> | Dataset              | ESGPT | ESGPT-C | FlexTPP | FlexTPP-C | Improvement |
> |----------------------|-------|---------|---------|-----------|-------------|
> | Type 2 diabetes mellitus | 0.648 | 0.649   | 0.641   | **0.638** | **0.010**   |
> | Transplanted kidney      | 0.690 | 0.697   | 0.670   | **0.668** | **0.022**   |
> | Transplanted lung        | 0.816 | 0.800   | 0.581   | **0.546** | **0.254**   |
>
> Median improvement: 0.10 nats, vs. std dev of 0.009 nats. The only case ESGPT outperforms is on Anemia (0.220 vs. 0.238).
>
> The mismatch between a Gaussian and arbitrary distributions becomes especially pronounced when generating longer sequences. Because the model is autoregressive—using its own outputs as inputs for future steps—errors from the incorrect Gaussian assumption compound over time. We confirm this on the event annotation task (Table 4):
>
> |                          | Full NLL (lower is better) | Mean AUC ROC (higher is better) |
> |--------------------------|----------------------------|----------------------------------|
> | FlexTPP (no access to input sequence) | -0.01 (0.003)               | 0.5 (0.0)                     |
> | Discrete Marks + avg duration | n/a                        | 0.90 (0.01)                     |
> | FlexTPP-C (ours)         | **-0.79** (0.006)          | **0.92** (0.01)                 |
> | ESGPT-C                  | -0.14 (0.13)               | 0.87 (0.01)                     |
>
> These results illustrate the critical benefit of modeling continuous marks expressively. While the data encoding is identical, our approach is both more general and performs better in practice. We will include ESGPT as a baseline in Tables 3 and 4 and cite the paper in the final version.
>
>
> > Difference to https://arxiv.org/abs/1909.12127
>
> Intensity-free Temporal Point Processes (IFTPP), like our FlexTPP, model the arrival time of the next event using a flexible one-dimensional distribution. We adopt their point process framework in Section 3.4, lines 96-98.
>
> However, the authors do not model more complex mark spaces such as variable-length and mixed-type.
>
> In addition, on the EasyTPP benchmark, **our approach outperforms IFTPP in all cases**.
>
>
> > https://arxiv.org/abs/1909.12127 [...] found that normalizing flows are not helpful
>
> The authors of that paper refer to particular expressive one-dimensional bijections that are expensive for sampling since they require numerically solving for an inverse (see their section 3.1). Neural spline flows [https://arxiv.org/abs/1906.04032] overcome this bottleneck since they are based on RQ splines, which have an analytic inverse function.
>
> > Regarding 4.2: How many patients are there per condition? How are you addressing the issue of multiple labels per patient inherent in EHR data? In Table 3 when you say "Lab tests w/o values" are you excluding the values only as prediction targets or as both prediction targets and features?
>
> 1. Between 64 and 1515 patients (average: ~700 patients), of which we use 70% for training, and 15% each for validation and testing.
> 2. We train one model for each condition, and filter the data for the relevant events: top 10 CPT-4 treatments for patients with that disease, top 5 lab tests for patients with that disease (see Appendix A.2)
> 3. We are also excluding the values as features. The left part of Table 3 evaluates Eq (13), which measures the quality of predicting treatments given an increasing number of longitudinal covariates (first column: only previous treatments, second column: first column + time and type of lab results, third column: second column + lab results, fourth column: third column + demographic information).
>
>
> > No notion of uncertainty or statistic soundness of many of the comparisons. This is especially important for small datasets like EHRSHOT.
>
> We report standard deviations in Appendix A, Tables 6, 9, 15, confirming significance. We will make the wording more precise on the significance of the difference between FlexTPP and FlexTPP-C in Table 3.
>
>
> > Ablation whether particular components (like the normalizing flows part) are helpful
>
> For now, please refer to the data presented in the EasyTPP benchmark in Table 1 of our paper (comparing the setup of IFTPP in https://arxiv.org/abs/1909.12127 which uses a log-normal mixture instead of our spline flow), as well as the new data on https://arxiv.org/abs/2306.11547 for the EHRSHOT data, which uses a regression head for continuous marks instead of the spline. We propose adding a dedicated ablation on the EasyTPP benchmark comparing autoregressive backbones (attention vs RNN) as well as one dimensional distributions (intensity vs regression vs mixture vs normalizing flow) to disentangle the contributions.
>
>
> > The conditional handling seems overly engineered. Why not just add it as an event at the start of the sequence?
>
> Indeed, we do that for the EHRSHOT data. But for feeding time series data, it is inefficient to have 10000 input timestamps each have its own token. The conditional network therefore preprocesses the data.
>
>
> We hope that this addresses the concerns of the reviewers and we are happy to address any further questions in the subsequent discussion.

---

> > ### Comment · Reviewer_ByqH · 2025-08-01
> >
> > Thank you for helping answer most of my concerns. I have updated my score to a weak reject.
> >
> > Line by line responses to items where I still have some concerns:
> >
> > > https://arxiv.org/abs/1909.12127 [...] found that normalizing flows are not helpful
> >
> > My concern is that 1909.12127 found an even bigger issue than sampling problems: they found that the performance (in terms of likelihood) of the two normalizing flow methods they investigated (DSFlow and SOSFlow) did not seem to meet the performance of the mixture model. See Figure 3.
> >
> > For your work, I have the same concern: that a simpler mixture model might perform better. I know you evaluate against IFTPP in some of your experiments, but that evaluation is complicated by the fact that you are comparing their mixture model with an RNN vs your normalizing flows with a more tuned transformer.
> >
> > > Difference to https://arxiv.org/abs/2306.11547
> >
> > Thank you for mostly addressing this concern. However, I just want to clarify:
> >
> > Did you explicitly check that the unimodal Guassian is the problem by only ablating that?
> >
> > What is the difference between ESGPT	and FlexTPP in that table? Is it only the Guassian vs normalizing flow, or is are there more underlying changes going on in this comparison?
> >
> > > Regarding 4.2
> >
> > Thanks for the reference to the appendix with confidence intervals, but I still have a lot of concerns.
> >
> > First, I cannot reproduce your patient counts. As a starting point, I was only able to find 47 patients for Cardiac transplant disorder which is outside your ranges.
> >
> > I used the following code to identify patients (using Athena to identify child concepts, and then searching EHRSHOT):
> >
> > > d_concepts = concept_ancestor.filter(pl.col('ancestor_concept_id') == 443575).select('descendant_concept_id').unique().rename({'descendant_concept_id': 'concept_id'})
> > >
> > > target_concepts = concept.join(d, on='concept_id')
> > > ehrshot.join(target_concepts.with_columns((pl.col('vocabulary_id') + '/' + pl.col('concept_code')).alias('code')), on='code').select('patient_id').unique()
> >
> > This is querying Athena (https://athena.ohdsi.org/search-terms/terms/320299, routing to https://athena.ohdsi.org/search-terms/terms/443575), with a final query of the following SNOMED codes:
> >  SNOMED/213151004,SNOMED/213152006,SNOMED/233844002,SNOMED/233933006,SNOMED/233934000,SNOMED/429257001,SNOMED/431186002,SNOMED/431896008,SNOMED/432773004,SNOMED/432843002,SNOMED/444855007,SNOMED/792842004
> >
> > 47 patients means you only have 7 positive patients in your test set.
> >
> > I'm also concerned that your error bars are not accurately representing the uncertainty here with a test set of only 7 positive patients. I think this might be a flaw with the standard deviation over repeated runs method of measuring uncertainty. I am concerned that simply resampling your splits could affect significantly affect results. Do your results drastically change when another 7 patients are picked?

---

> > > ### Author Response · Authors · 2025-08-04
> > >
> > > Thanks for your prompt response, we appreciate your interest and constructive feedback.
> > >
> > > > For your work, I have the same concern: that a simpler mixture model might perform better.
> > >
> > > Let’s test this directly. We replaced the spline in our model with a log-normal mixture, as used in IFTPP. The table below shows the test negate log-likelihoods (lower is better):
> > >
> > > | Method                      | Amazon           | Retweet          | Taxi             | Taobao           | StackOverflow    |
> > > |----------------------------|------------------|------------------|------------------|------------------|------------------|
> > > | IFTPP                      | -0.496 (0.002)   | 10.344 (0.016)   | -0.453 (0.002)   | -1.318 (0.017)   | 2.233 (0.009)    |
> > > | FlexTPP w/ log-normal mix. | 2.210 (0.018)    | 6.449 (0.001)    | -0.734 (0.002)   | -1.355 (0.006)   | 2.253 (0.005)    |
> > > | FlexTPP w/ spline (ours)   | **-0.633** (0.039) | **5.646** (0.070) | **-0.763** (0.005) | **-1.402** (0.013) | **2.133** (0.004) |
> > >
> > >
> > > **The spline always performs better than the log-normal mixture**, with differences exceeding standard deviations over five runs. We tuned the number of mixture components for each dataset on the validation data.
> > >
> > > Regarding simplicity, the spline and the Gaussian have comparable execution times for likelihoods and sampling. And existing implementations (e.g. TensorFlow Probability and normalizing flow libraries, Neural Spline Flow repo) make splines easy to use.
> > >
> > > We will add this ablation to the paper.
> > >
> > >
> > > > Did you explicitly check that the unimodal Guassian is the problem by only ablating that? What is the difference between ESGPT and FlexTPP in that table? Is it only the Guassian vs normalizing flow, or is are there more underlying changes going on in this comparison?
> > >
> > > Yes, no other changes. We use our code and only replace the continuous mark heads with a Gaussian instead of the spline.
> > >
> > >
> > > > [Regarding 4.2] First, I cannot reproduce your patient counts. As a starting point, I was only able to find 47 patients for Cardiac transplant disorder which is outside your ranges.
> > >
> > > We have been using SNOMED code “SNOMED/233932001” for Cardiac, resulting in 64 patients.
> > >
> > > Here is the full list of SNOMED codes we used for disease datasets:
> > >
> > > | Dataset | Code |
> > > |-----------------------------|------------------------|
> > > | Type 2 diabetes mellitus | SNOMED/313436004 |
> > > | Transplanted kidney | SNOMED/737295003 |
> > > | Transplanted lung | SNOMED/737296002 |
> > > | Dyspnea | SNOMED/267036007 |
> > > | Atrial fibrillation | SNOMED/49436004 |
> > > | Cardiac transplant disorder | SNOMED/233932001 |
> > > | End-stage renal disease | SNOMED/46177005 |
> > > | Transplanted heart | SNOMED/739024006 |
> > > | Congestive heart failure | SNOMED/42343007 |
> > > | Chronic pain | SNOMED/82423001 |
> > > | Neoplasm of female breast | SNOMED/93796005 |
> > > | Obstructive sleep apnea | SNOMED/78275009 |
> > > | Diabetes with complication | SNOMED/44054006 |
> > > | Anemia | SNOMED/271737000 |
> > > | Coronary artery disease | SNOMED/451041000124103 |
> > > | Hypothyroidism | SNOMED/40930008 |
> > > | Acute myeloid leukemia | SNOMED/91861009 |
> > > | Depressive disorder | SNOMED/35489007 |
> > > | Transplanted liver | SNOMED/737297006 |
> > > | Acute kidney injury | SNOMED/39104002 |
> > >
> > >
> > > > Do your results drastically change when another 7 patients are picked?
> > >
> > > No, our results are consistent across different random splits.
> > >
> > > We evaluated performance over random permutations of the patients with multiple seeds (0 to 7), and our method consistently outperforms both ESGPT and ESGPT-C. FlexTPP-C is better than FlexTPP in all but one seed.
> > >
> > > Here are the results from these seeds (lower is better):
> > >
> > > | Seed | ESGPT | ESGPT-C | FlexTPP | FlexTPP-C |
> > > |------|-------|---------|---------|-----------|
> > > | 0 | 0.534 | 0.522 | 0.405 | **0.394** |
> > > | 1 | 0.537 | 0.523 | 0.423 | **0.406** |
> > > | 2 | 0.549 | 0.568 | **0.412** | 0.422 |
> > > | 3 | 0.520 | 0.488 | 0.406 | **0.376** |
> > > | 4 | 0.555 | 0.524 | 0.409 | **0.378** |
> > > | 5 | 0.580 | 0.521 | 0.443 | **0.414** |
> > > | 6 | 0.511 | 0.511 | 0.394 | **0.381** |
> > > | 7 | 0.532 | 0.513 | 0.434 | **0.382** |
> > >
> > > The results shown in Table 3 of the paper correspond to seed 42.
> > >
> > > We hope this addresses the reviewer's concerns, we look forward to further comments.

---

> > > > ### Comment · Reviewer_ByqH · 2025-08-04
> > > >
> > > > Thank you for the detailed response. I have updated my score to an accept.
> > > >
> > > > Thank you for explicitly comparing against the relevant prior work, in particular showing the downsides of the guassian and mixture distribution (from IFTPP). It's very interesting that IFTPP's finding that log normals were better than normalizing flows doesn't hold up. I would be curious if that's due to the change in backbone (to a transformer) or the change in normalizing flow implementation methods.
> > > >
> > > > Looking back at the Cardiac Transplant, I can now reproduce your numbers. The reason I got confused is because SNOMED/233932001 is an invalid, now outdated SNOMED code, and I didn't think EHRSHOT had it. As a minor note, I think your query isn't quite ideal, as you should probably query both the invalid SNOMED/233932001 and the newer replacement of SNOMED/429257001, which gets you 69 patients. But it seems to have a minimal impact.

---

### Official Review · Reviewer_ZDCH · 2025-06-29

**Clarity:** 3
**Significance:** 2
**Originality:** 2
**Rating:** 5
**Confidence:** 2

**Summary:**

This paper extends Marked Temporal Point Processes (MTPPs) to support variable-length marks with mixed discrete and continuous data types, proposing FLEXTPP—a Transformer-based architecture that flattens heterogeneous event data into sequences and models them autoregressively. The approach uses categorical distributions for discrete marks and normalizing flows for continuous values, for distribution modeling.

**Questions:**

To the best of my understanding, the proposed approach proceeds as follows:
(1) Start with events consisting of timestamps and mixed-type marks.
(2) Flatten each event into a sequence of the form $[t_1,  type_1,  mark_{1,1}, \dots, mark_{1,n}, t_2, type_2, \dots]$.
(3) Feed this sequence into a Transformer.
(4) Train or sample the model in an autoregressive manner.

Can you specially list what has been done more than above, and what's the contribution/innovation from a point process view.

I may be misunderstanding some aspects and would welcome clarification to reassess my evaluation of this work.

**Ethical Concerns:**

["NO or VERY MINOR ethics concerns only"]

**Final Justification:**

The author clearly listed their contributions in the rebuttal. I have also read through other reviews and the authors' responses. I am convinced it is a solid paper.

**Limitations:**

Yes, the paper discussed their limitations

**Quality:**

3

**Strengths And Weaknesses:**

**Strengths:**

The paper addresses a practical limitation in existing MTPP methods by supporting variable-length, mixed-type marks, which is important for real-world applications such as electronic health records. The proposed framework is clean and general, and the empirical evaluation is relatively comprehensive, showing good performance across multiple benchmarks.

**Weaknesses:**
(1) The model conditions each mark component on all previous times and marks in a fully autoregressive manner, which can be unnecessarily restrictive, or misrepresent the data. For example, blood pressure and heart rate are recorded at the same time for a patient, but there is no natural dependence on each other.

(2) While the approach is well-executed, it largely applies standard sequence modeling to flattened temporal data. I feel the main novelty is in the data representation (mixed-type, variable-length marks) rather than methodological innovation.

(3) The likelihood values are reported in the experiments, but not sure how significance the improvement is without statistical test.

**Suggestion.**
The paper complicates its presentation by deriving the approach through intensity functions in Section 3.1, despite ultimately proposing an intensity-free model that directly parameterizes $p(t_i, m_i | H_{t_i})$. The detour through intensity functions $\lambda(t, m |H_t)$ and the conversion in Equation (2) is unnecessary. Equation (3) merely states the standard likelihood for time series data, not a derived result. The exposition would be clearer and more focused if it began with the goal of modeling $p(t_i, m_i | H_{t_i})$ directly, with only a brief mention of intensity-based methods as a contrasting approach.

---

> ### Author Rebuttal · Authors · 2025-07-31
>
> We thank the reviewer for the helpful comments. We particularly appreciate the call to list contributions from a point process point of view.
>
> > The model conditions each mark component on all previous times and marks in a fully autoregressive manner, which can be unnecessarily restrictive or misrepresent the data. For example, blood pressure and heart rate are recorded at the same time for a patient, but there is no natural dependence on each other.
>
> The described knowledge about the true dependent structure can be integrated into our model: If there is prior knowledge about the (in)dependence between certain predictions, this can be guaranteed to be fulfilled by the model by masking out the corresponding attention weights in the transformer.
>
> For example, assuming we want to predict $p(\text{blood pressure}, \text{heart rate}|\text{past events})$ as independent variables, we can zero out entries of the fully autoregressive attention mask:
>
> $\\begin{pmatrix} 1 & 0 \\\\ 1 & 1 \\end{pmatrix} \\mapsto \\begin{pmatrix} 1 & 0 \\\\ 0 & 1 \\end{pmatrix} $
>
> This results in an exactly independent prediction $p(\text{blood pressure}, \text{heart rate}|\text{past events}) = p(\text{blood pressure}|\text{past events}) p(\text{heart rate}|\text{past events})$.
>
> > While the approach is well-executed, it largely applies standard sequence modeling to flattened temporal data. [...] Can you specially list what has been done more than above, and what's the contribution/innovation from a point process view.
>
> From a point-process view, we contribute:
>
> **A new state-of-the-art for intensity-free modeling**.
>
> Contrary to the original IFTPP paper, we find that normalizing flows are a highly effective basis for intensity-free modeling (see their comment on normalizing flows in section 3.1 in https://arxiv.org/pdf/1909.12127). We combine a transformer backbone with neural spline flows, achieving closed-form likelihoods and fast sampling. This outperforms their approach (Table 1).
>
>
> **Intensity-free outperforms intensity-based**.
>
> Previous work argued that intensity-based methods dominate due to "fewer modeling restrictions" [https://arxiv.org/pdf/2412.19634, Section 4]. Our approach brings intensity-free modeling back on the table, outperforming all intensity-based methods (except for DLHP in one dataset, see Table 1), all while avoiding numeric integration at training and rejection sampling at inference time.
>
> From a high level, this strengthens the connection between the point process and generative modeling literatures.
>
>
> **Point processes go beyond homogeneous marks**.
>
> We show that point processes are a natural model for generative models of series of complex events such as patient records and time series annotations.
>
> [Note that reviewer ByqH pointed us to related work [1], which encodes data in a similar way to our model. However, their model regresses continuous marks instead of modeling them with a universal distribution. This is not expressive enough and empirically deteriorates under autoregressive sampling.]
>
>
> **Blackbox methods allow for interpretation**. [Thanks to suggestion by reviewer hdCQ]
>
> Despite being a blackbox method, our model allows extracting information about the underlying point process. In detail, we replicated an experiment from [2] by successfully extracting a triggering kernel from our model. Our extraction closely follows the ground truth compared to extracting the relation between events from the attention weights, which is the baseline in that paper.
>
>
> Again thanks for pushing us to think more about the point process perspective. We propose overhauling the list of contributions in our introduction section.
>
>
> - [1] Matthew B et al. "Event Stream GPT: A Data Pre-processing and Modeling Library for Generative, Pre-trained Transformers over Continuous-time Sequences of Complex Events", NeurIPS 2023.
> - [2] Isik et al., "Hawkes Process with Flexible Triggering Kernels", MLHC 2023.
>
>
> > The likelihood values are reported in the experiments, but not sure how significance the improvement is without statistical test.
>
> We report standard deviations in Appendix A, Tables 6, 9, 15, confirming significance. We will make the wording more precise on the significance of the difference between FlexTPP and FlexTPP-C in Table 3.
>
>
> We hope that this clarifies the concerns of the reviewers. We are happy to provide further details in the subsequent discussion.

---

> > ### Comment · Reviewer_ZDCH · 2025-08-06
> >
> > Thank you for clearly states your contributions. I have increased my score.

---

### Official Review · Reviewer_M7Vi · 2025-07-03

**Clarity:** 3
**Significance:** 4
**Originality:** 3
**Rating:** 5
**Confidence:** 4

**Summary:**

The paper proposes a marked temporal point process framework for predicting event type, event time, and variable-length discrete/continuous longitudinal covariates, given event history and static covariates, by leveraging an autoregressive likelihood. It employs an autoregressive transformer for encoding event history and conditioning on static covariates. Additionally, the paper leverages normalizing flows for event time and continuous longitudinal covariates prediction. Experimental results demonstrate that the proposed  approach achieves competitive event time and event type likelihood predictions compared to baselines on two real-world and one synthetic dataset.

**Questions:**

- Could you provide a complexity analysis in terms of the number of parameters, training/inference time, and scalability relative to baselines?
- Eqn 9: Could you clarify how the normalizing flow function is specified, *e.g.*, what is different or shared across marker types and event times? Also what is $f^\prime$?
- Could you clarify the empirical benefits of conditioning on longitudinal covariates, *i.e*., Equation 10 *vs.* Equation 12?
- Tables 3 and 4: (i) Could you clarify the models used for "only discrete events" and "Lab tests w/o values"? (ii) Could you provide complete results, including baselines from Table 2?
- Figure 3: The qualitative results in Figure 3 are difficult to follow; could you clarify?
- How does the model capture the influence of event interactions on event time?

**Ethical Concerns:**

["NO or VERY MINOR ethics concerns only"]

**Final Justification:**

Thanks to such a comprehensive rebuttal that addresses most of my concerns, I have increased my score to acceptance.

**Limitations:**

- The main limitations of the proposed approach are its complexity relative to baselines and its lack of interpretability. The paper should expand the discussion on these aspects.

**Quality:**

3

**Strengths And Weaknesses:**

**Strengths**

- The paper is relatively well-written and easy to follow.
- The use of autoregressive transformers for conditional generation given variable length discrete/continuous longitudinal covariates is interesting.
- Experimental results demonstrate that the proposed approach achieves competetive ikelihood predictions compared to baselines.

**Weaknesses**
- The proposed approach seems more complex and might not be scalable relative to baselines, which is worsened by the use of normalizing flows for both continuous marker metadata and event time.
- The paper seems to focus on the likelihood metric, overlooking model interpretability, which is important in practice. For example, by not modeling the intensity function, it is unclear how the model captures the influence of event interactions on event time. The paper should also benchmark the interpretability aspect and compare it with classical Hawkes and interpretable neural Hawkes approaches, *e.g.*, [1, 2].
- The proposed approach seems to yield small gains relative to baselines (including FlexTPP vs. FlexTPP-C). The paper should also report error bars of the reported negative likelihood.
- While the paper clearly describes the ablation study of the proposed model (Equations 10-12), the experimental results do not seem to highlight the benefits of conditioning on the variable length discrete/continous longitudinal covariates, which is the key contribution of this work compared to baselines. The paper should also consistently provide benchmarks for the proposed FlexTPP (with/without the longitudinal covariates), *i.e.*, Eqn 10 *vs.* Eqn 12.
- Table 4: The paper should report the metrics on event type and event time predictions separately and provide non-likelihood-based metrics, such as RMSE, for event time predictions.
- *Minor*: Line 91 - The intensity probabilities should be conditioned on the event history.

**References**

- [1] Isik et al., "Hawkes Process with Flexible Triggering Kernels",  MLHC 2023.
- [2] Shixiang et al. , "Neural spectral marked point processes",  ICLR 2022.

---

> ### Author Rebuttal · Authors · 2025-07-31
>
> We thank the reviewer for the detailed, diverse, and helpful comments.
>
> We begin with main limitations, followed by specific responses.
>
> > The main limitations [...] are its complexity relative to baselines and its lack of interpretability.
>
> Let’s start with interpretability:
>
> > The paper should also benchmark the interpretability aspect and compare it with classical Hawkes and interpretable neural Hawkes approaches, e.g., [1, 2].
>
> That is an excellent suggestion! Building on the reviewer’s references [1, 2], we find that our model offers a comparable level of interpretability—albeit in a local, query-based form—by targeted inspection of how past events influence future intensities.
>
> **Interpretability through targeted intensity inspection**
>
> In Hawkes Process with Flexible Triggering Kernels [1], the method learns a kernel $q_{m_1 m_2}(|t_1 - t_2|)$. It measures how an event $(m_1, t_1)$ influences the intensity of a future event $(m_2, t_2)$.
>
> Similarly, Neural Spectral Marked Point Processes [2] learns kernels $q(m_1, t_1, m_2, t_2)$ that compose the intensity for the next event.
>
> We can extract these kernels using our Eq. (2, left), which computes intensities from the event pdf and cdf.
>
> We experimentally confirm that our model accurately predicts intensities such as  $\\lambda(t_2|t_1)$ via Eq. (2, left) for a synthetic example from [1, Figure 2]. We find that the RMSE distance between the true kernel and the one extracted from our method is $0.01$. You can imagine Figure 2 in reference [1] with our line closely following the true kernel.
>
> We will add the above result as an interpretability section to our paper, and are happy to provide any additional details to the reviewer upon request.
>
> **Global vs. Local Interpretability**
>
> Taking a step back, there is an inherent tradeoff between globally interpretable models and highly flexible ones.
>
> Structured approaches that learn kernel-based pairwise event influences allow for a simple complete understanding of the model’s behavior. However, they are limited in flexibility and can misrepresent data with nonlinear dynamics. For example, in marketing, an effect such as ad fatigue can’t be modeled, where seeing an ad once or twice increases the chance of conversion, but repeated exposures eventually stop having any effect or may even cause disengagement. The ground truth structure of real data is usually not clear and so interpreting a kernel-based model can yield spurious explanations.
>
> While our model is not globally interpretable, it supports local interpretability through queries about how specific past events affect future intensities, allowing fine-grained analysis and uncertainty estimation.
>
>
> > [Regarding Complexity] The proposed approach seems more complex and might not be scalable relative to baselines, which is worsened by the use of normalizing flows [...].
>
> The flow adds negligible cost (one millionth of a second on consumer GPU); most computation comes from the autoregressive transformer, similar to SAHP and NHP
>
>
> > Eqn 9: Could you clarify how the normalizing flow function is specified, e.g., what is different or shared across marker types and event times? Also what is f’?
>
> The normalizing flow models 1D conditional distributions over continuous values, with shared structure over types. Just like categorical logits, it is parameterized based on the history through the transformer.
>
> Eqn (9) is the standard change of variables equation underlying normalizing flows and autoregressive models in particular. Such a model learns an invertible function mapping $x$ to a standard normal in the latent space. The density $p(x)$ implied by the flow is proportional to the likelihood of $z=f(x)$ under the standard normal, and proportional to the volume change $f’(x)$, which in 1D is just the derivative of $df(x)/dx$ with respect to its input.
>
>
> > How does the model capture the influence of event interactions on event time?
>
> The autoregressive transformer takes in all previous events (times, mixed-type marks) as a vector $H_i$ to predict the parameters of the next event time $p(t_{i}|H_i)$.
>
> This is similar to Self-Attentive Hawkes Processes and Transformer Hawkes Processes, although they follow an intensity-based approach, which requires numeric integration at training and rejection sampling at inference time. Additionally, we outperform them on EasyTPP in Table 1. We model the next event time as a one-dimensional probability distribution in an intensity-free fashion, which avoids these issues.
>
>
> > Could you provide a complexity analysis in terms of the number of parameters, training/inference time, and scalability relative to baselines?
>
> **Our model is somewhat larger, but trains and runs at comparable speed** to competitors.
>
> In detail, let us report the geometric mean for the requested metrics:
>
> - Parameter count: We use a mean of 325k parameters, which is about 3-10x larger competitors.
> - Training speed: We achieve a training speed of around 2400 sequences of length 200 per second, over 2x faster than AttNHP and NHP. RMTPP, SAHP, THP and IFTPP are 2-5x faster than us.
> - Generation speed: ~70 sequences of length 200 per second, with similar relation to competitors as training. We did not implement a KV-Cache, which speeds up sampling by a factor in sequence length.
> - Scalability: Self-attention inherently has access to the entire event history in each prediction. This makes memory scale linearly $O(n)$ with sequence length (one would naively assume that memory use is quadratic, but the attention matrix never has to be realized in memory, e.g., using FlashAttention [3]). Notably, we can fit a sequence of over a million entries on a GPU with 12GB of VRAM for our largest model. Also, transformers with four orders of magnitude more parameters than our biggest model have been deployed on edge devices (e.g., Llama 3-8B on a Raspberry Pi at 500ms/token using quantization [4]).
>
> References:
>
> - [3] Dao. FlashAttention-2: Faster Attention with Better Parallelism and Work Partitioning. ICLR 2024
> - [4] Ardakani et al. LLMPi: Optimizing LLMs for High-Throughput on Raspberry Pi. CVPR Workshop Paper 2025
>
>
> > Could you clarify the empirical benefits of conditioning on longitudinal covariates, i.e., Equation 10 vs. Equation 12?
>
> Eq. 10 and 12 are not directly comparable since they do not concern the same variables. To assess whether including additional longitudinal variables is useful, we need to look at Eq. (13). It tells us the likelihood of events (e.g. medical interventions for health data) depending on the knowledge of the covariates: $p(t_\text{procedure}, t_\text{procedure}| \text{past procedures}, \text{covariates})$.
>
> Table 3 shows the NLL as in Eq. (13), e.g. the first row (lower is better, std < 0.002 in all cases):
>
> | Covariates | Procedure NLL |
> |--|--|
> | No covariates | 0.691 |
> | + time, lab test type | 0.202 |
> | + lab test results | 0.147 |
> | + demographics | **0.143** |
>
> The more longitudinal covariates we include, the better we can predict the next medical procedure.
>
>
> > The proposed approach seems to yield small gains relative to baselines (including FlexTPP vs. FlexTPP-C). The paper should also report error bars of the reported negative likelihood.
>
> We report standard deviations in Appendix A, Tables 6, 9, 15, confirming significance.
>
> For FlexTPP (our model, unconditional; Eq. 11) vs FlexTPP-C (our model, conditional; Eq. 12), the reviewer is correct that only 60% of the winning NLL is at least one standard deviation away from the other model.
>
>
> > Figure 3: The qualitative results in Figure 3 are difficult to follow; could you clarify?
>
> Figure 3 shows a sequence from the time series annotation dataset, where the goal is to infer events affecting the input signal. The ground truth events (blue) consist of a frequency change by a factor of 1.14, and additional noise of 1.63.
>
> The discrete-only model (orange) predicts event start times and types but cannot predict durations or continuous parameters. The depicted duration is chosen to fit the text. In contrast, our model (green) predicts start, end, and parameter values, accurately modeling the full event structure. To our knowledge, this is the first point process model applied to such a task.
>
>
> > Table 4: The paper should report the metrics on event type and event time predictions separately and provide non-likelihood-based metrics, such as RMSE, for event time predictions.
>
> We agree that non-likelihood-based metrics can be helpful for interpreting the model performance. Thus, we reported AUC ROC in Table 4.
>
> Here are the RMSE values for Table 4 (standard deviations in brackets):
> | Model | RMSE |
> |---|--|
> | no condition | 0.198 (0.001) |
> | only discrete marks | 0.192 (0.009) |
> | FlexTPP-C | **0.152** (0.002) |
>
> While this clearly favors our method, we think that RMSE is not a good metric for this task. A model that samples events autoregressively can have a good RMSE but if one of the generations fails (e.g. because the model is tuned for RMSE via a regression loss), the rest of the sequence cannot be meaningful.
>
> We therefore report AUC ROC in the paper since (a) they are based on a full generated sequence of events from each model and (b) are an accepted quantity in the time series annotation literature [5].
>
> [5] Schmidl et al.: Anomaly detection in time series: a comprehensive evaluation. VLDB 2022.

---

### Official Review · Reviewer_hdCQ · 2025-07-05

**Clarity:** 4
**Significance:** 2
**Originality:** 3
**Rating:** 5
**Confidence:** 5

**Summary:**

This paper extends Marked Temporal Point Processes (MTPPs) by introducing a framework that supports mixed-type (discrete/continuous) and variable-length marks through normalizing flows, particularly effective in healthcare scenarios like EHR analysis. It establishes conditional modeling as a core principle using an autoregressive Transformer architecture, which eliminates numerical integration bottlenecks and enhances dependency capture compared to RNN-based model. The proposed intensity-free approach employs direct joint probability modeling, significantly improving computational efficiency in complex event generation. Empirical validation across medical and financial domains demonstrates superior prediction accuracy, especially in long-term forecasting through advanced historical encoding, revealing synergistic interactions between temporal dynamics and mark information.

**Questions:**

Please clarify whether complexity calculations exclude attention weight matrix generation and provide quantitative GPU memory usage data, as the self-attention mechanism inherently requires O(N²).

**Ethical Concerns:**

["NO or VERY MINOR ethics concerns only"]

**Limitations:**

yes

**Paper Formatting Concerns:**

No formatting concerns are found.

**Quality:**

3

**Strengths And Weaknesses:**

Strengths：
This study demonstrates significant advancements in marked temporal point process modeling through two core innovations：1）it pioneers hybrid-type (discrete+continuous) and variable-length mark processing using normalizing flows, overcoming the fixed-type mark limitations of traditional MTPPs like the RMTPP framework；2) the conditional modeling framework via autoregressive Transformer，achieving higher conditional information utilization compared to the attention-based improvements, particularly enhancing structured output tasks in cross-modal scenarios.
Weaknesses：
The research exhibits certain limitations: The combined computational load from Transformer and normalizing flow architectures increases parameter count by 5× compared to RNN-based models, potentially hindering edge device deployment. While effective for medium-dimensional marks like medical codes, the approach lacks validation on high-dimensional sparse marks (e.g., genomic data).

---

> ### Author Rebuttal · Authors · 2025-07-31
>
> We thank the reviewer for their helpful feedback!
>
> > This study demonstrates significant advancements in marked temporal point process modeling through two core innovations [hybrid-type processes and conditional information utilization]
>
> Thanks for acknowledging these contributions. We would also like to point out these additional contributions of our work:
>
> 1. **Intensity-free outperforms intensity-based**. Previous work argued that intensity-based methods dominate due to "fewer modeling restrictions" [https://arxiv.org/pdf/2412.19634, Section 4]. Our approach brings intensity-free modeling back on the table, outperforming all intensity-based methods (except for DLHP in one dataset, see Table 1), all while avoiding numeric integration at training and rejection sampling at inference time.
> 2. **Successful extraction of triggering kernels** (new thanks to reviewer M7Vi). We are adding a successful evaluation on the interpretability of our model to the camera ready version. See our answer to M7Vi for details.
>
>
> > The combined computational load from Transformer and normalizing flow architectures increases parameter count by 5× compared to RNN-based models, potentially hindering edge device deployment.
>
> Our model is several orders of magnitude smaller than transformers that have been deployed to edge devices. For example, recent work quantizes the weights of Llama 3-8B (8 billion parameters, which is four orders of magnitude larger than our largest model) to run it on a Raspberry Pi 5 at ~500ms/token [1].
>
> The normalizing flow itself has at most 88(!) parameters predicted by the transformer, and likelihood computation/sampling takes around one millionth of a second on a consumer GPU for each entry. This is comparable to the compute required for a Hawkes process intensity evaluation.
>
>
> [1] Ardakani et al. LLMPi: Optimizing LLMs for High-Throughput on Raspberry Pi. CVPR Workshop Paper 2025
>
>
> > Please clarify whether complexity calculations exclude attention weight matrix generation and provide quantitative GPU memory usage data, as the self-attention mechanism inherently requires $O(N^2)$.
>
> Transformers are not memory-bound: This remarkable effect can be achieved by computing the attention matrix in chunks and never storing it in full in memory. Then, the **memory consumption of a transformer is linear in sequence length** $O(n)$. Computing the attention matrix as needed without ever realizing it in full is one of the tricks in modern attention implementations such as PyTorch’s `scaled_dot_product_attention`, e.g. based on FlashAttention [2].
>
> Concretely, on a GPU with 12GB VRAM and our largest EasyTPP model, **we can fit a maximum sequence length of two million tokens** for a batch size of 1.
>
> [2] Dao. FlashAttention-2: Faster Attention with Better Parallelism and Work Partitioning. ICLR 2024
>
>
> We hope that this elucidates the properties and contributions of our approach, and we are happy to address further questions in the upcoming discussion.

---

### Public Comment · ~Guanglin_Zhou2 · 2025-11-11
**The code repo is unavailable**

Nice paper, authors. I was wondering when the code repo will be publicly available? Thanks!

---

> ### Public Comment · ~Felix_Draxler1 · 2025-11-19
>
> Hi Guanglin,
>
> thanks! We are still pending an internal code review, unfortunately. Will keep you posted!
>
> Best,
> Felix

---

> > ### Public Comment · ~Guanglin_Zhou2 · 2025-11-23
> >
> > Thanks, Felix. Looking forward to the code.

---

> > > ### Public Comment · ~Felix_Draxler1 · 2026-02-07
> > > **Finally, code is out**
> > >
> > > Dear Guanglin Zhou, the code is now available at https://github.com/czi-ai/FlexTPP. Thanks for your patience

---

### Decision · Program_Chairs · 2025-09-17

**Decision:**

Accept (spotlight)

**Comment:**

Following the final round of reviews, the paper received four "Accept" recommendations. The reviewers recognized the technical novelty of the proposed approach for modeling marked temporal point processes, as well as its strong empirical performance. They particularly emphasized the method’s ability to handle variable-length and mixed-type marks, which they considered a notable strength. Additionally, the rebuttal was well-received and had a positive influence on the reviewers’ assessments, as it effectively addressed most of their initial concerns.